# TAKE A STEP BACK: EVOKING REASONING VIA ABSTRACTION IN LARGE LANGUAGE MODELS

**Huaixiu Steven Zheng**[*]    **Swaroop Mishra**[*]    **Xinyun Chen**    **Heng-Tze Cheng**
**Ed H. Chi**    **Quoc V Le**    **Denny Zhou**
Google DeepMind

## ABSTRACT

We present STEP-BACK PROMPTING, a simple prompting technique that enables LLMs to do abstractions to derive high-level concepts and first principles from instances containing specific details. Using the concepts and principles to guide reasoning, LLMs significantly improve their abilities in following a correct reasoning path towards the solution. We conduct experiments of STEP-BACK PROMPTING with PaLM-2L, GPT-4 and Llama2-70B models, and observe substantial performance gains on various challenging reasoning-intensive tasks including STEM, Knowledge QA, and Multi-Hop Reasoning. For instance, STEP-BACK PROMPTING improves PaLM-2L performance on MMLU (Physics and Chemistry) by 7% and 11% respectively, TimeQA by 27%, and MuSiQue by 7%.

*The purpose of abstraction is not to be vague, but to create a new semantic level in which one can be absolutely precise. — Edsger W. Dijkstra*

## 1 INTRODUCTION

The field of natural language processing (NLP) is witnessing a ground-breaking revolution because of the Transformer-based (Vaswani et al., 2017) large language models (LLMs) (Devlin et al., 2018; Raffel et al., 2020; Brown et al., 2020; Anil et al., 2023). Scaling up the model size and pre-training corpus (Hoffmann et al., 2022; Chowdhery et al., 2022) has brought remarkable improvement in model capabilities and sample efficiency with insights from the scaling law (Kaplan et al., 2020; Hoffmann et al., 2022), as well as emergent abilities (Wei et al., 2022a) such as multi-step reasoning (Wei et al., 2022b; Zhou et al., 2022) and instruction following (Mishra et al., 2022b; Wei et al., 2021).

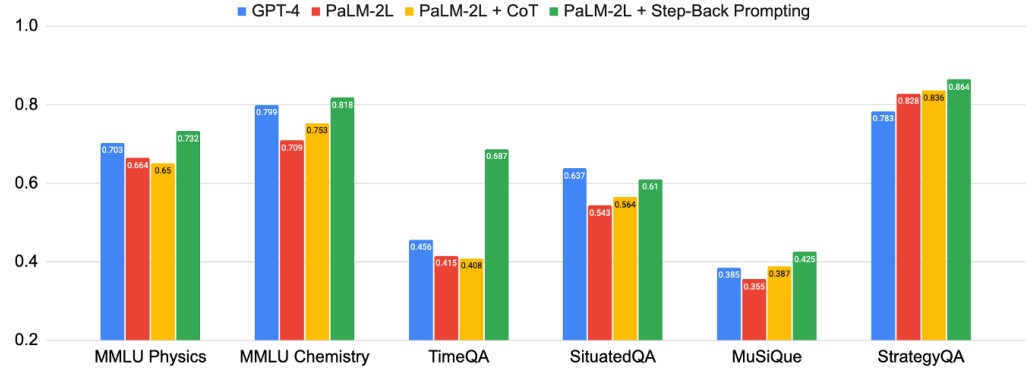

Figure 1: Strong Performance of STEP-BACK PROMPTING: our proposed Abstraction-and-Reasoning scheme leads to a substantial improvement in a wide range of challenging tasks in STEM, Knowledge QA and Multi-Hop Reasoning requiring complex (often multi-hop) reasoning.

---

[*]Equal Contribution

Despite the great advancements, complex multi-step reasoning remains challenging for even the state-of-the-art LLMs. Lightman et al. (2023) show that process-supervision with step-by-step verification is a promising remedy to improve the correctness of intermediate reasoning steps. Techniques such as Chain-of-Thought (Wei et al., 2022b) were introduced to produce a coherent series of intermediate reasoning steps to increase the success rate of following the right decoding path. Inspired by the fact that when faced with challenging tasks humans often step back and do abstractions to arrive at high-level principles to guide the process, we propose STEP-BACK PROMPTING to ground reasoning on abstractions to reduce the chance of making errors in the intermediate reasoning steps.

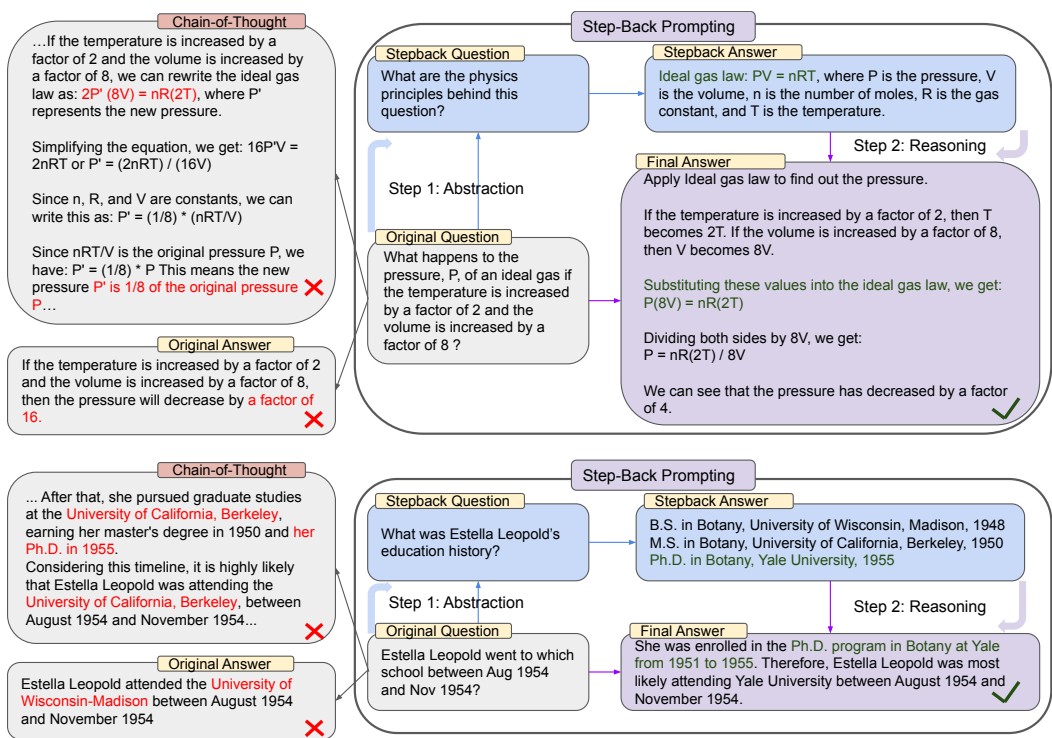

Figure 2: Illustration of STEP-BACK PROMPTING with two steps of Abstraction and Reasoning guided by concepts and principles. *Top*: an example of MMLU high-school physics (Hendrycks et al., 2020) where the first principle of Ideal Gas Law is retrieved via abstraction. *Bottom*: an example from TimeQA (Chen et al., 2021) where the high-level concept of education history is a result of the abstraction. *Left*: PaLM-2L (Anil et al., 2023) fails to answer the original question. Chain-of-Thought prompting (Wei et al., 2022b; Kojima et al., 2022) ran into errors during intermediate reasoning steps (highlighted as red). *Right*: PaLM-2L (Anil et al., 2023) successfully answers the question via STEP-BACK PROMPTING.

Among many of the cognitive skills, abstraction (Lachmy et al., 2022) is ubiquitous to humans' ability to process vast amounts of information and derive general principles. For example, Kepler compressed thousands of measurements into Kepler's three laws of planetary motion, which precisely describe the orbits of planets around the Sun (Russell, 1964). In critical decision-making, humans find abstraction to be helpful since it provides a broader view of the environment. This work explores how LLMs can tackle complex tasks involving many low-level details through a two-step process of abstraction-and-reasoning. The first step is to show LLMs how to step back through in-context learning – prompting them to derive high-level abstractions such as concepts and principles for a specific example. The second step is to leverage the reasoning ability to reason on top of the high-level concepts and principles. We use few-shot exemplar demonstrations to execute STEP-BACK PROMPTING on LLMs.

We experiment across a range of tasks involving domain specific reasoning such as Physics and Chemistry, knowledge-intensive question answering requiring factual knowledge, multi-hop commonsense reasoning. We observe significant performance improvements (up to 27%) in PaLM-2L (Anil et al.,

2023) demonstrating the efficacy of STEP-BACK PROMPTING in tackling complex tasks, which are otherwise challenging due to the amount of details needed for reasoning. Figure 1 shows a summary of all the key results presented in this paper. Some the tasks are very challenging: both PaLM-2L and GPT-4 achieve only $\sim 40\%$ accuracy on TimeQA and MuSiQue. Chain-of-Thought prompting leads to a minor improvement on a few tasks, while STEP-BACK PROMPTING improves the performance of PaLM-2L across the board: $7\%$ and $11\%$ on MMLU Physics and Chemistry, $27\%$ on TimeQA, and $7\%$ on MuSiQue.

We conduct a variety of analyses and find that STEP-BACK PROMPTING leads to strong performance improvements (up to $36\%$) over chain-of-thought (CoT) prompting (Wei et al., 2022b) and "take-a-deep-breath" (TDB) prompting (Yang et al., 2023). We perform a qualitative evaluation where we find that Step-Back fixes a large portion of errors of the base model (up to $\sim 40\%$) while introducing a small portion of new errors (max $\sim 12\%$). We also conduct an error analysis and find that majority of the errors made by STEP-BACK PROMPTING is attributed to the intrinsic limitations of reasoning capabilities of LLMs while abstraction skills are relatively easy to demonstrate to LLMs, pointing out the direction for future improvements of methods alike STEP-BACK PROMPTING.

## 2 STEP-BACK PROMPTING

STEP-BACK PROMPTING is motivated by the observation that many tasks contain a lot of details, and it is hard for LLMs to retrieve relevant facts to tackle the task. As shown in the first example (top) in Figure 2, for a Physics question of "*What happens to the pressure, P, of an ideal gas if the temperature is increased by a factor of 2 and the volume is increased by a factor of 8 ?*", the LLM can deviate from the first principle of Ideal Gas Law when reasoning directly on the question. Similarly, a question of "*Estella Leopold went to which school between Aug 1954 and Nov 1954?*" is very hard to address directly given the detailed time range constraint. In both cases, asking a step-back question helps the model to solve the problem effectively.

We define a step-back question as a derived question from the original question at a higher level of abstraction. For instance, instead of directly asking "*which school Estella Leopold went to during a specific period*", a step-back question (Figure 2 bottom) would ask about the "*education history*", which is a high-level concept encompasses the original question. Answering the step-back question of "*Estella Leopold's education history*" in this case will provide all the necessary information to reason about "*which school Estella Leopold went to during a specific period*". The premise is that the step-back question is typically much easier. Grounding the reasoning on top of such abstractions helps to avoid reasoning errors in the intermediate steps such as the example shown in Figure 2 (left) from Chain-of-Thought. In short, STEP-BACK PROMPTING consists two simple steps:

- **Abstraction**: Instead of addressing the question directly, we first prompt the LLM to ask a generic step-back question about a higher-level concept or principle, and retrieve relevant facts about the high-level concept or principle. The step-back question is unique for each task in order to retrieve the most relevant facts.
- **Reasoning**: Grounded on the facts regarding the high-level concept or principle, the LLM can reason about the solution to the original question. We term this as *Abstraction-grounded Reasoning*.

In the following sections, we present an empirical study of STEP-BACK PROMPTING on a range of challenging tasks covering STEM, Knowledge QA, and Multi-Hop Reasoning involving complex reasoning.

## 3 EXPERIMENTAL SETUP

Here we define the tasks and models we experiment with. We also describe our evaluation metric and the baselines we consider.

### 3.1 TASKS

We experiment with the following diverse tasks: (a) STEM, (b) Knowledge QA, and (c) Multi-Hop Reasoning. We describe below the datasets we consider (see Appendix B for more details).

- **STEM**: We evaluate MMLU and GSM8K for STEM tasks. MMLU (Hendrycks et al., 2020) contains a series of benchmarks across diverse domains to evaluate the model's language understanding. We consider the high school physics and chemistry portions of MMLU because of the deep reasoning involved.

- **Knowledge QA**: We consider TimeQA (Chen et al., 2021) since it contains complex queries that require challenging time-sensitive knowledge. We also experiment with SituatedQA (Zhang & Choi, 2021), another challenging open-retrieval QA dataset requiring the model to answer questions given temporal or geographical contexts.

- **Multi-Hop Reasoning**: We experiment with MuSiQue (Trivedi et al., 2022), a hard multi-hop reasoning dataset created via composable pairs of single-hop questions, and StrategyQA (Geva et al., 2021) with open-domain questions that demand some strategy to solve.

## 3.2 MODELS

We use the following state-of-the-art LLMs: instruction-tuned PaLM-2L (Anil et al., 2023), GPT-4 (OpenAI, 2023), and Llama2-70B (Touvron et al., 2023).

## 3.3 EVALUATION

Conventional evaluation metrics such as accuracy, F1 score have limitations specifically for evaluating the generations of state-of-the-art LLMs since these models often generate long-form answers which are hard to capture. We instead conduct an evaluation using the PaLM-2L model where we few-shot prompt the model to identify equivalence between target answers and the model predictions. Few-shot examples, prompts and other details used for this evaluation are in Appendix C.

## 3.4 BASELINE METHODS

- **PaLM-2L, PaLM-2L 1-shot**: PaLM-2L is either queried directly with the question or has a single demonstration exemplar of question-answer included in the prompt.

- **PaLM-2L + CoT, PaLM-2L + CoT 1-shot**: PaLM-2L model is queried with zero-shot CoT prompting (Kojima et al., 2022): "*Let's think step by step*" is appended to the question. For 1-shot, One demonstration example of a question and answer pair is provided in the prompt, where the answer is in the style of CoT (Wei et al., 2022b).

- **PaLM-2L + TDB**: Zero-shot prompting with "*Take a deep breath and work on this problem step-by-step.*" (Yang et al., 2023) prepended to the question.

- **PaLM-2L + RAG**: For Sections 5 and 6, we use retrieval-augmented generation (RAG) where the retrieved passage is used as context by the LLM.

- **GPT-4 and Llama2-70B**: we run GPT-4 and Llama2-70B on MMLU tasks for all methods. In addition, we also run GPT-4 on all baselines for all tasks.

We do not use RAG for STEM tasks, because of the inherent reasoning nature of the tasks contrary to the other fact-seeking datasets. All inferences are done using greedy decoding.

# 4 STEM

We evaluate STEP-BACK PROMPTING on STEM tasks (Hendrycks et al., 2020) to gauge the efficacy of our method on reasoning in highly specialized domains. We explain below our experimental setup, result, and analysis of applying STEP-BACK PROMPTING on the MMLU high-school Physics and Chemistry, and GSM8K benchmarks.

## 4.1 STEP-BACK PROMPTING

Questions in the MMLU benchmarks require deeper reasoning. Furthermore, they also require understanding and application of formulae which are often physics and chemistry principles and concepts. In this case, we first demonstrate to the model abstraction skills in the form of concepts and first principles such as *Newton's first law of motion*, *Doppler effect*, and *Gibbs free energy* etc.

Table 1: Strong performance of STEP-BACK PROMPTING on MMLU tasks across three model families. CoT: zero-shot Chain of Thought prompting (Kojima et al., 2022), TDB: Take a Deep Breath prompting (Yang et al., 2023).

| Method | MMLU Physics | MMLU Chemistry |
|---|---|---|
| PaLM-2L | 66.4% (0.8%) | 70.9% (0.9%) |
| PaLM-2L 1-shot | 64% (1.6%) | 75.6% (0.4%) |
| PaLM-2L + CoT | 65% (2%) | 75.3% (1.5%) |
| PaLM-2L + CoT 1-shot | 61.5% (1.8%) | 76.6% (1%) |
| PaLM-2L + TDB | 65.7% (0.7%) | 73.8% (1.1%) |
| PaLM-2L + Step-Back (ours) | **73.2%** (1.9%) | **81.8%** (1.4%) |
| GPT-4 | 69.4% (2.0%) | 80.9% (0.7%) |
| GPT-4 1-shot | 78.4% (2.4%) | 80.5% (1.6%) |
| GPT-4 + CoT | 82.9% (0.5%) | 85.3% (1.0%) |
| GPT-4 + CoT 1-shot | 79.3% (1.0%) | 82.8% (0.5%) |
| GPT-4 + TDB | 74.4% (4.0%) | 81.5% (1.3%) |
| GPT-4 + Step-Back (ours) | **84.5%** (1.2%) | **85.6%** (1.4%) |
| Llama2-70B | 51.9% (3.6%) | 55.7% (2.1%) |
| Llama2-70B 1-shot | 57.3% (1.6%) | 58.5% (2.5%) |
| Llama2-70B + CoT | 59.3% (2.0%) | 64.1% (1.2%) |
| Llama2-70B + CoT 1-shot | 59.6% (2.0%) | **68.1%** (1.4%) |
| Llama2-70B + TDB | 60.4% (2.1%) | 63.6% (1.9%) |
| Llama2-70B + Step-Back (ours) | **64.8%** (1.5%) | 66.7% (1.6%) |

The implicit step-back question here is "*what are the physics or chemistry principles and concepts involved in solving this task?*". We provide demonstrations to the model to recite the relevant principles for solving the task from its own knowledge (see Appendix D.1 for few-shot exemplars).

## 4.2 RESULTS

Table 1 illustrates model performance across various setups across three model families: PaLM-2L, GPT-4, and Llama2-70B. Average accuracy over 5 evaluation runs is reported along with standard deviations (in the parentheses). PaLM-2L baseline performance is 66.4% and 70.9% on Physics and Chemistry, respectively. We find that CoT and TDB zero-shot prompting do not significantly increase model performance, which could be due to the inherent difficulty and deep reasoning associated with these tasks. PaLM-2L 1-shot and PaLM-2L + CoT 1-shot do not improve against the baseline much either, highlighting the challenge of demonstrating the reasoning steps to the model. In contrast, STEP-BACK PROMPTING significantly improves model performance: +7% and +11% compared to PaLM-2L. Similarly, with GPT-

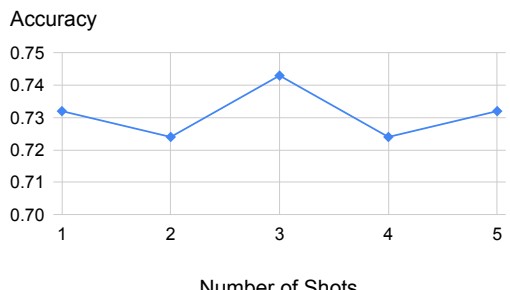

Figure 3: Ablation study of STEP-BACK PROMPTING accuracy using PaLM-2L on MMLU high-school Physics against the number of few shot exemplars: robust performance with respect to a varying number of shots.

4 and Llama2-70B models, STEP-BACK PROMPTING is very competitive among all the baseline methods we tested, showing that STEP-BACK PROMPTING is model-agnostic. We present the results of GSM8K in Appendix A.1.

## 4.3 ABLATION AND ANALYSIS

**Few-shot Ablation**: First, in Figure 3, we observe that STEP-BACK PROMPTING is robust to the number of few-shot exemplars of (question, principles) pairs used as demonstrations. Adding more

Table 2: Strong performance of STEP-BACK PROMPTING on Knowledge QA tasks. CoT: Chain of Thought prompting, TDB: Take a Deep Breath prompting, RAG: retrieval-augmented generation. STEP-BACK PROMPTING results in significant performance improvements.

| Method | TimeQA | TQA Easy | TQA Hard | SituatedQA |
|---|---|---|---|---|
| PaLM-2L | 41.5% | 42.6% | 40.4% | 54.3% (0.3%) |
| PaLM-2L 1-shot | 40.7% | 41.7% | 39.1% | 51.8% (0.6%) |
| PaLM-2L + CoT | 40.8% | 41.8% | 39.8% | 56.4% (0.2%) |
| PaLM-2L + CoT 1-shot | 38.1% | 39.3% | 36.8% | 54% (0.8%) |
| PaLM-2L + TDB | 40.9% | 42.6% | 39.1% | 54% (0.5%) |
| PaLM-2L + RAG | 57.4% | 67.8% | 46.8% | 59.3% (0.4%) |
| PaLM-2L + Step-Back (ours) | 66% | 70.4% | 61.6% | 57.5% (0.3%) |
| PaLM-2L + Step-Back + RAG (ours) | **68.7%** | **75.2%** | **62.3%** | 61% (0.4%) |
| GPT-4 | 45.6% | 48.9% | 42.6% | **63.2%** (0.4%) |

demonstration examples beyond a single example does not lead to further improvements. This indicates that the task of retrieving the relevant principles and concepts is relatively easy through in-context learning and a single demonstration suffices. Therefore, we use a single exemplar for few-shot prompting throughout the paper except the ablation studies.

**Error Analysis**: Comparing the predictions of STEP-BACK PROMPTING to the baseline PaLM-2L model for MMLU high-school Physics: we find that STEP-BACK PROMPTING corrects $20.5\%$ errors from the baseline while introducing $11.9\%$ errors.

To further understand where the errors come from in STEP-BACK PROMPTING, we annotate all the wrong predictions of STEP-BACK PROMPTING in the test set, and categorize them into 5 classes (see Appendix E.1 for examples in each class):

- **Principle Error**: The error happens at the step of Abstraction, where the first principles generated by models are wrong or incomplete.
- **Factual Error**: There is at least one factual error when the model recites its own factual knowledge
- **Math Error**: There is at least one math error in the intermediate steps when math calculations are involved in deriving the final answer.
- **Context Loss**: There is at least one error where the model response loses context from the question, and deviates from addressing the original question
- **Reasoning Error**: We define Reasoning Error as when the model makes at least one error in the intermediate Reasoning steps before arriving at the final answer.

All five types of errors are happening during the Reasoning step except *Principle Error* which points to the failure of the Abstraction step. As shown in Figure 4 (right), *Principle Error* comprises only a small fraction of the errors the model makes: more than 90% of the errors happen at the Reasoning step. Among the four error types during Reasoning, *Reasoning Error* and *Math Error* are the major error categories. This corroborates with the finding in the ablation study above that very few exemplars are needed to demonstrate to LLMs the Abstraction skill. Reasoning step is still the bottleneck of how well STEP-BACK PROMPTING can perform tasks such as MMLU requiring complex reasoning. For MMLU Physics specifically, the Reasoning and Math skills are critical for solv-

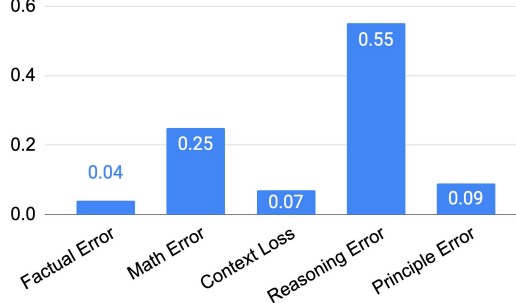

Figure 4: Error Analysis of STEP-BACK PROMPT-ING on MMLU high-school Physics: five classes of errors Step-Back makes with Reasoning being the dominating class.

ing the problems successfully: even if the first principles are retrieved correctly, deep reasoning and math are involved to derive a correct final answer through a typical multi-step reasoning process.

## 5 KNOWLEDGE QA

We evaluate STEP-BACK PROMPTING on question-answering benchmarks requiring intensive factual knowledge. Knowledge QA has been challenging for LLMs. In this section, we first describe the experimental setup, followed by results and analysis on STEP-BACK PROMPTING.

### 5.1 STEP-BACK PROMPTING

We evaluate STEP-BACK PROMPTING on TimeQA (Chen et al., 2021) and SituatedQA (Zhang & Choi, 2021) in the Knowledge QA category. We first show the LLMs how to do Abstraction through in-context demonstrations. The step-back question "*What was Estella Leopold's education history*" in Figure 2 is generated by the LLM through few-shot demonstrations (see Appendix D.2 for details). Given the knowledge-intensive nature of these queries, we use retrieval augmentation (RAG) in combination with STEP-BACK PROMPTING. The step-back question is used to retrieve relevant facts, which work as additional context (see Table 14 for the prompt) to ground the final reasoning step.

### 5.2 RESULTS

We evaluate the models on the test set of TimeQA. As shown in Table 2, the baseline models of GPT-4 and PaLM-2L achieved $45.6\%$ and $41.5\%$, highlighting the difficulty of the task. Applying either CoT or TDB zero-shot (and one-shot) prompting to the baseline model shows no improvement. In contrast, augmenting the baseline model by regular retrieval augmentation (RAG) improves the accuracy to $57.4\%$, highlighting the fact-intensive nature of the task. The result of Step-Back + RAG shows the effectiveness of going back to a high-level concept, which enables much more reliable retrieval augmentation: the accuracy on TimeQA achieves a remarkable $68.7\%$.

Next, we segment TimeQA into the Easy and Hard difficulty levels provided in the original dataset. As expected, all methods perform worse on the Hard subset. While RAG can improve the Easy accuracy from $42.6\%$ to $67.8\%$, the improvement is much smaller on the Hard accuracy: $40.4\%$ to $46.8\%$. This is where STEP-BACK PROMPTING shines by retrieving facts regarding high-level concepts to ground the final reasoning: Step-Back + RAG further improves the Hard accuracy to $62.3\%$, outperforming GPT-4's $42.6\%$ from GPT-4. We hypothesize that facts regarding the high-level concepts (such as *education history*) are much more accessible than the low-level details.

On the SituatedQA benchmark, we observe a moderate quality gain from $54.3\%$ to our best method of Step-Back + RAG ($61\%$) with a small gap to GPT-4's $63.2\%$. Similar to TimeQA, prompting techniques such as CoT and TDB don't help significantly for SituatedQA.

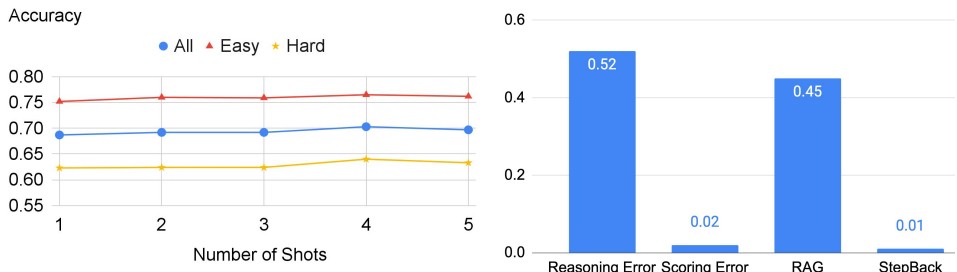

Figure 5: Ablation and error analysis of STEP-BACK PROMPTING on TimeQA. *Left*: ablation against the number of few-shot exemplars. *Right*: four classes of errors Step-Back makes with Reasoning and RAG being the dominant error sources.

### 5.3 ABLATION AND ANALYSIS

**Few-shot Ablation**: We observe in Figure 5 (left) that the performance of STEP-BACK PROMPTING on TimeQA is robust to the number of exemplars used in demonstration, highlighting again the sample efficiency of in-context learning Abstraction skills for models like PaLM-2L.

Table 3: Results of STEP-BACK PROMPTING on Multi-Hop Reasoning. CoT: Chain of Thought prompting, TDB: Take a Deep Breath prompting, RAG: retrieval augmentation generation. The average accuracy is over 5 evaluation runs with the standard deviations included in the parentheses.

| Method | MuSiQue | StrategyQA |
|---|---|---|
| PaLM-2L | 35.5% (3%) | 82.8% (0.7%) |
| PaLM-2L 1-shot | 29.0% (0.5%) | 76.6% (0.5%) |
| PaLM-2L + CoT | 38.7% (3.2%) | 83.6% (0.4%) |
| PaLM-2L + CoT 1-shot | 38.5% (2.2%) | 76.8% (1.4%) |
| PaLM-2L + TDB | 39.0% (2.3%) | 82.7% (0.9%) |
| PaLM-2L + RAG | 39.6% (2.8%) | 84.2% (0.5%) |
| PaLM-2L + Step-Back (ours) | 42.6% (3.1%) | 82.7% (0.4%) |
| PaLM-2L + Step-Back + RAG (ours) | **42.8%** (2.0%) | **86.4%** (1%) |
| GPT-4 | 38.5% (0.2%) | 78.3% (1.1%) |

**Error Analysis:** Figure 5 (right) shows the breakdown of all the remaining errors made by STEP-BACK PROMPTING on TimeQA. Similar to Section 4.3, we categorize the errors into

- **StepBack**: The step-back question generated is not helpful in solving the task.
- **RAG**: RAG fails to retrieve relevant information despite that the step-back question is on target.
- **Scoring Error**: The evaluation by the judge model made a mistake.
- **Reasoning Error**: The retrieved context is relevant, but the model still fails to reason through the context to arrive at the right answer.

We find that the StepBack rarely fails. In contrast, we find more than half of the errors are due to reasoning errors. Additionally, $45\%$ of errors are due to failure in retrieving the right information despite that Abstraction provided by step-back makes it a much easier task. This reflects the difficulty level of the TimeQA task. Additional error analysis of TimeQA is in Appendix A.

## 6 MULTI-HOP REASONING

We evaluate STEP-BACK PROMPTING on challenging Multi-Hop reasoning benchmark MuSiQue (Trivedi et al., 2022) and StrategyQA (Geva et al., 2021). We follow the same protocol as Section 5 to implement STEP-BACK PROMPTING.

Table 3 shows performance of various baselines on the dev set of MuSiQue and StrategyQA. Baseline performance of PaLM-2L and GPT-4 are low ($35.5\%$ and $38.5\%$ for PaLM-2L and GPT-4 respectively) in MuSiQue since it is a hard multihop reasoning benchmark. In contrast, StrategyQA has stronger baselines ($82.8\%$ and $78.3\%$ for PaLM-2L and GPT-4 respectively) probably because it is a binary classification task. CoT and TDB improve model performance a bit in the case of MuSiQue ($\sim 3\%$ and $3.5\%$ respectively) which can be attributed to the inherent reasoning nature of this task where these methods are shown to be helpful. In the case of StrategyQA, there is no significant performance gain with CoT and TDB which could be due to the high baseline performance in this task, with limited scope for these prompting methods to improve performance. Often, 1-shot performance is significantly lower than their zero-shot methods, which could be attributed to potential example bias (Zhao et al., 2021; Parmar et al., 2023). RAG improves model performance ($\sim 4\%$ and $2\%$ for MuSiQue and StrategyQA respectively.) STEP-BACK PROMPTING with the power of abstraction produces the best performance of all methods: $42.8\%$ in MuSiQue and $86.4\%$ in StrategyQA, significantly outperforming GPT-4 on both tasks. We present a detailed error analysis on StrategyQA in Appendix A.3.

## 7 DISCUSSION

Abstraction helps humans to solve complex tasks by removing irrelevant details and distilling high-level concepts and principles to guide the problem-solving process. STEP-BACK PROMPTING breaks

complex tasks such as knowledge-intensive QA, multi-hop reasoning, and science questions into two separate steps of Abstraction and Reasoning. We demonstrate through empirical experiments that Abstraction is an easy skill for the LLMs such as PaLM-2L via sample-efficient in-context learning. Grounding on the high-level concepts and principles, LLMs can leverage their intrinsic Reasoning capabilities to derive the solution. This reduces the chance of reasoning failures in the intermediate steps and is shown to improve the performance on a wide range of complex reasoning tasks. Despite the success, through error analysis, we find that Reasoning is still one of the hardest skills for LLMs to acquire: it is still the dominant failure mode even after the large reduction of task complexity by STEP-BACK PROMPTING.

Nevertheless, Abstraction is neither necessary nor possible in all scenarios. For instance, the task can be as simple as *who was the president of the United States in 2000?*, in which case there is no such need to step back and ask a high-level question as the answer to such questions is readily available. Questions such as *what is the speed of light?* point to the first principles themselves. Doing Abstraction in this case would not make a difference either.

## 8 RELATED WORK

### 8.1 PROMPTING

Few-shot prompting (Brown et al., 2020; Liu et al., 2023; Mishra et al., 2022a; Wei et al., 2022b) has significantly improved model performance across a range of tasks without requiring updating any model parameters. Our work STEP-BACK PROMPTING is in the same category as the chain-of-thought prompting (Wei et al., 2022b) and scratchpad (Nye et al., 2021) owing to its simplicity and generic nature. But our approach is focused on the key idea of abstraction which is inspired from the fact that taking a step back often helps humans in performing complex tasks. Our work is also related to the recitation-augmented language models (Sun et al., 2022); however in contrast to their work, we explicitly perform step-back and abstraction, with optional use of retrieval augmentation depending on the nature of the task at hand.

### 8.2 DECOMPOSITION

Decomposing a task into simpler tasks and solving these tasks to complete the original task has been an effective way (Zhou et al., 2022; Patel et al., 2022; Khot et al., 2022; Press et al., 2022) to improve model performance on complex tasks. Several prompting methods have been successful in this regard. Our work STEP-BACK PROMPTING, in contrast, is on making the question more abstract and high-level, which is different from decomposition that is often a low-level breakdowns of the original question. For instance, a generic question for *which employer did Steve Jobs work for in 1990?* could be *what is the employment history of Steve Jobs?* While decomposition would lead to sub-questions such as *What was Steve Jobs doing in 1990?*, *Was Steve Jobs employed in 1990?* and *If Steve Jobs was employed, who was his employer?* Furthermore, abstract questions such as *what is the employment history of Steve Jobs?* are often generic in nature to have a many-to-one mapping since many questions (e.g. *which employer did Steve Jobs work for in 1990?* and *which employer did Steve Jobs work for in 2000?*) can have the same abstract question. This is in contrast to decomposition where there is often a one-to-many mapping since there are multiple decomposed sub-problems necessary to solve a given question.

## 9 CONCLUSION

We introduce STEP-BACK PROMPTING as a simple yet generic method to elicit deep reasoning via abstraction in large language models. Experimentation on LLMs across fact-seeking, commonsense reasoning and domain-specific reasoning benchmarks shows that STEP-BACK PROMPTING significantly improves model performance. We hypothesize that abstraction helps models to hallucinate less and reason better, probably reflecting the true nature of the model which are often hidden while responding to the original question without abstraction. We hope our work will inspire more human-inspired approaches to elicit the hidden potential of large language models.

ACKNOWLEDGEMENT

We thank Andrew Dai, Adams Yu and Hannah Rashkin of Google DeepMind for their insightful feedback on this paper.

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

Table 4: Step-Back Prompting on GSM8K. CoT: zero-shot Chain of Thought prompting (Kojima et al., 2022), TDB: Take a Deep Breath prompting (Yang et al., 2023), DP: Decomposed Prompting (Khot et al., 2022). The Table reports the average accuracy over 5 evaluation runs, with standard deviations in the parentheses.

| Method | GSM8K |
|---|---|
| PaLM-2L | 75.8% (0.2%) |
| PaLM-2L 1-shot | **84.5%** (0.4%) |
| PaLM-2L + CoT | **84.4%** (0.2%) |
| PaLM-2L + CoT 1-shot | 81% (0.2%) |
| PaLM-2L + TDB | 82.2% (0.2%) |
| PaLM-2L + DP | 82.2% (0.08%) |
| PaLM-2L + Step-Back (ours) | **84.3%** (0.2%) |

## A GSM8K RESULTS, AND ERROR ANALYSIS

### A.1 GSM8K RESULTS

We present in Table 4 the results of STEP-BACK PROMPTING on GSM8K along with other strong baselines from PaLM-2L runs. We observe that STEP-BACK PROMPTING achieved competitive performance together with zero-shot CoT and 1-shot standard prompting. We hypothesize that the simplicity of principles (e.g. addition, subtraction, etc.) in GSM8K makes it not absolutely necessary to retrieve the principles first before reasoning. Nonetheless, we still find that STEP-BACK PROMPTING is the most competitive among all the prompting methods we tested, including the "Take a Deep Breath" prompting optimized for GSM8K in Yang et al. (2023) and Decomposed Prompting in Khot et al. (2022).

### A.2 TIMEQA ERROR ANALYSIS

We conduct error analysis to understand where STEP-BACK PROMPTING fixes the errors the baseline models make. Figure 6 shows that compared to the predictions of baseline PaLM-2L, STEP-BACK PROMPTING can fix 39.9% of the predictions where the baseline prediction is wrong, while causing 5.6% errors.Furthermore, Step-Back + RAG fixes 21.6% errors coming from RAG. The % of errors introduced by STEP-BACK PROMPTING to RAG is still relatively low (6.3%). Together, this shows that the STEP-BACK PROMPTING is helpful most of the time, signifying the need and effectiveness of doing Abstraction before directly addressing the original question.

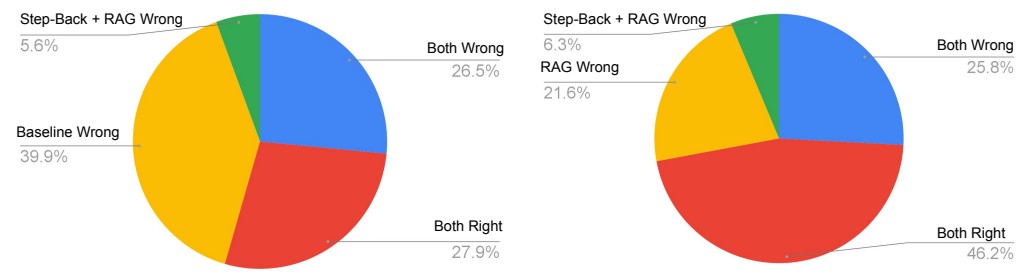

Figure 6: Error Analysis of Step-Back Prompting on TimeQA. *Left*: Step-Back + RAG vs Baseline predictions. *Right*: Step-Back RAG vs RAG predictions. Step-Back + RAG can fix 39.9% of the predictions where the baseline prediction is wrong while causing 5.6% errors. Furthermore, Step-Back + RAG fixes 21.6% errors coming from RAG. The % of errors introduced by STEP-BACK PROMPTING to RAG is still relatively low (6.3%).

Table 5: Stats of the evaluation datasets used in this paper.

| Domain | Dataset | Split | Number of Examples |
|---|---|---|---|
| STEM | MMLU high-school Physics | Test | 151 |
| | MMLU high-school Chemistry | Test | 203 |
| | GSM8K | Test | 1319 |
| Knowledge QA | TimeQA | Test | 5226 |
| | TimeQA Easy | Test | 2613 |
| | TimeQA Hard | Test | 2613 |
| | SituatedQA | Test | 2901 |
| Multi-hop Reasoning | MuSiQue | Dev | 2417 |
| | StrategyQA | Dev | 229 |

### A.3 STRATEGYQA ERROR ANALYSIS

Figure 7 shows the error analysis of StrategyQA on the predictions of Step-Back + RAG against the baseline model and the raw retrieval augmentation variant of PaLM-2L. Compared to the baseline, Step-Back + RAG can turn $15.4\%$ wrong predictions into correct predictions, while leading to $6.1\%$ errors the other way around. Furthermore, Step-Back + RAG fixes $12.7\%$ errors coming from RAG. The errors introduced to RAG by Step-Back are just $4.4\%$.

## B DATASET DETAILS

Table 5 shows the split and number of examples used for evaluations in TimeQA, StrategyQA, MMLU, and GSM8K.

## C EVALUATION DETAILS

### C.1 FEW-SHOT EXAMPLES FOR EVALUATION WITH PALM-2L

Given the model free-form outputs and the target label, we use one positive and one negative output as few-shot examples to demonstrate to the scoring model how to score the output. Table 6 illustrates the prompt we used for the scoring model. We parse out the "Yes" or "No" answer from the scoring model output as a TRUE or FALSE score of the model output.

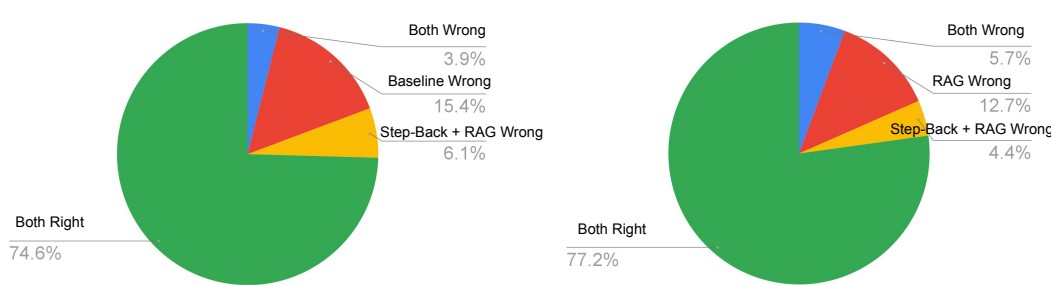

Figure 7: Error Analysis of Step-Back Prompting on StrategyQA. *Left*: Step-Back + RAG vs Baseline predictions. *Right*: Step-Back + RAG vs RAG predictions. Step-Back + RAG is able to turn $15.4\%$ wrong predictions into correct predictions, while leading to $6.1\%$ errors the other way around. Furthermore, Step-Back + RAG fixes $12.7\%$ errors coming from RAG. The errors introduced to RAG by Step-Back are just $4.4\%$.

Are the following two answers to the given question equivalent? Do not consider whether the answers are right or wrong, but only whether they are equivalent. Directly state "Yes" or "No".
**Question**: Which title was conferred to Anna Muzychuk in 2007?
**Answer 1**: Anna Muzychuk was conferred the title of International Master (IM) in 2007. She earned the title by scoring three norms in rapid chess tournaments.
**Answer 2**: International Master
**Answer 1 (short)**: International Master
**Answer 2 (short)**: International Master
**Are the two answers equivalent?** Yes
**Question**: What state is Seattle located in?
**Answer 1**: Seattle is in Washington State.
**Answer 2**: The answer is George Washington.
**Answer 1 (short)**: Washington State
**Answer 2 (short)**: George Washington
**Are the two answers equivalent?** No
**Question**: <Question>
**Answer 1**: <Model Output>
**Answer 2**: <Target Label>

Table 6: Illustration of few shot evaluation with the PaLM-2L model.

MMLU Physics/Chemistry First-Principle Prompt

You are an expert at Physics/Chemistry. You are given a Physics/Chemistry problem. Your task is to extract the Physics/Chemistry concepts and principles involved in solving the problem. Here are a few examples:

Question: <Question Example1>
Principles Involved: <Principles Example1>
...
Question: <Question Example5>
Principles Involved: <Principles Example5>
Question: <Question>
Principles Involved:

Table 7: Prompt of extracting the underlying principles involved in MMLU physics and chemistry questions.

## C.2 HYPER-PARAMETERS FOR EVALUATION WITH PALM-2L

We use PaLM-2L as the scoring model for evaluation. We experiment with different sampling temperatures, and find that $T = 1$ gives us a highly-accurate evaluation. For example, we sampled 100 test examples and the model predictions, and manually rated the correctness of the model scoring. We found that out of 4 trials, the model scoring agrees with human ratings $97\%$, $98\%$, $99\%$ and $99\%$ of the time.

## D PROMPTS AND FEW SHOT EXAMPLES

### D.1 STEM

For MMLU high-school Physics and Chemistry, we first prompt the model to generate the first principles behind the question. Using the generated first principles, we further prompt the model to generate the final answer through few-shot demonstrations The prompt generating first principles is shown in Table 7 for MMLU high-school Physics and Chemistry.

| MMLU Physics/Chemistry Final Answer Prompt |
| --- |
| You are an expert at Physics/Chemistry. You are given a Physics/Chemistry problem and a set of principles involved in solving the problem. Solve the problem step by step by following the principles. Here are a few examples:

Question: \<Question Example1\>
Principles: \<Principles Example1\>
Answer: \<Answer Example1\>
...
Question: \<Question Example5\>
Principles: \<Principles Example5\>
Answer: \<Answer Example5\>
Question: \<Question\>
Principles: \<Principles\>
Answer: |

Table 8: Prompt of querying the model for final answer with first principles behind the question in MMLU high-school Physics and Chemistry.

After extracting the first principles of solving a particular question, we formulate the prompt in Table 8 to query the model for the final answer.

Tables 9-10 show one demonstration exemplar of Question-Principles-Answer triplets for MMLU high-school Physics and Chemistry, respectively. For GSM8K, given the simplicity of the principles, we directly combine the principles and the solution in the demonstration exemplar. Table 11 shows the exemplar we used in the paper.

### D.2 KNOWLEDGE QA

We use the following prompting in Table 12 to demonstrate to the LLM on asking a step-back question for TimeQA and SituatedQA including up to 5 exemplar demonstrations of pairs of Original Question and Step-back Question.

Table 13 shows 5 exemplars from the Train split of TimeQA and SituatedQA as demonstrations of asking step-back questions.

The step-back question is extracted from the model output using the prompt. Using the step-back question, we do retrieval augmentation. Using both the retrieval augmentations from the original question and the step-back question, we formulate the final prompt to query the model for the final answer, as shown in Table 14.

### D.3 MULTI-HOP REASONING

For Multi-Hop Reasoning, we use the same prompting template as in Knowledge QA to ask the step-back question, and query for the final answer given the retrieval augmentations. Table 15 shows 5 demonstration exemplars for asking step-back questions from the Train split of MuSiQue and StrategyQA.

### D.4 BASELINE PROMPTS

For standard zero-shot and few-shot prompting of the baseline model, we formulate the prompt using the template in Table 16 with up to 1 exemplars.

### D.5 CHAIN OF THOUGHT (COT)

For zero-shot CoT prompting, we simply append *Let's think step by step.* to the question to query the model.

| | |
|---|---|
| Question | A spherical conductor carries a net charge. How is this charge distributed on the sphere?
0) The charge is evenly distributed on the surface.
1) The charge resides on the surface only; the distribution of charge on the surface depends on what other charged objects are near the sphere.
2) The charge moves continually within the sphere.
3) The charge is distributed uniformly throughout the sphere. |
| Principles | Coulomb's Law: the force between two charged particles is proportional to the product of their charges and inversely proportional to the square of the distance between them, $F = k * q1 * q2 / r$, where F is the electric force, k is a constant, q1 and q2 are the charges the particles carry, and r is the distance between them. |
| Answer | Using the Principles of Coulomb's Law, we can solve the problem as following:
Step 1: Apply Coulomb's Law to find out how charges are distributed on the surface.
In the case of a spherical conductor, the charges on the surface will repel each other. The further apart the charges are, the less force they will exert on each other. Therefore, the charges will distribute themselves evenly on the surface of the sphere, as this is the configuration that minimizes the repulsive force between them.

Step 2: Apply Coulomb's Law to find out what happens if there are other charges present.
The distribution of charge on the surface may also be affected by the presence of other charged objects near the sphere. For example, if a negatively charged object is brought near a positively charged sphere, the negative charges on the sphere will be repelled and will move to the opposite side of the sphere. This will result in a non-uniform distribution of charge on the surface of the sphere.

Therefore, the correct answer is 1) The charge resides on the surface only; the distribution of charge on the surface depends on what other charged objects are near the sphere. |

Table 9: Few-shot demonstration exemplar for MMLU high-school Physics.

| Question | A sample of an unknown chloride compound was dissolved in water, and then titrated with excess Pb(NO3)2 to create a precipitate. After drying, it is determined there are 0.0050 mol of precipitate present. What mass of chloride is present in the original sample?
0) 0.177 g
1) 0.355 g
2) 0.522 g
3) 0.710 g |
|---|---|
| Principles | Precipitation reactions: Precipitation reactions occur when two soluble salts are mixed and form an insoluble product, called a precipitate. The precipitate can be separated from the solution by filtration or centrifugation.
Molar mass: The molar mass of a substance is the mass of one mole of that substance. The molar mass is expressed in grams per mole (g/mol).
Limiting reactant: The limiting reactant is the reactant that is completely consumed in a chemical reaction. The amount of product formed is determined by the amount of limiting reactant. |
| Answer | Assuming the unknown chloride compound is MCl, where M represents the metal cation, the balanced chemical equation for the precipitation reaction is:
$Pb(NO_3)_2(aq) + 2MCl(aq) \longrightarrow PbCl_2(s) + 2MNO_3(aq)$

Since $Pb(NO_3)_2$ is in excess, MCl is the limiting reactant. The stoichiometry of the reaction indicates that 2 moles of MCl produce 1 mole of $PbCl_2$ precipitate. Therefore, 0.0050 mol of $PbCl_2$ corresponds to 0.010 mol of MCl.

The mass of chloride in the original sample can be calculated using the molar mass of chloride (35.45 g/mol):
0.010 mol Cl × 35.45 g/mol = 0.355 g Cl

The correct answer is 1) 0.355 g. |

Table 10: Few-shot demonstration exemplar for MMLU high-school Chemistry.

| Question | Bella bought stamps at the post office. Some of the stamps had a snowflake design, some had a truck design, and some had a rose design. Bella bought 11 snowflake stamps. She bought 9 more truck stamps than snowflake stamps, and 13 fewer rose stamps than truck stamps. How many stamps did Bella buy in all? |
|---|---|
| Answer | Let us find and apply the math principles to solve the problem step by step:

Step 1. Addition: Calculate the number of truck stamps.
Bella bought 11 snowflake stamps. She bought 9 more truck stamps than snowflake stamps: there are 11 + 9 = 20 truck stamps.

Step 2. Subtraction: Calculate the number of rose stamps.
Bella bought 13 fewer rose stamps than truck stamps: there are 20 - 13 = 7 rose stamps.

Step 3. Addition: Calculate the total number of stamps in all three colors.
Bella bought 11 snowflake stamps, 20 truck stamps, 7 rose stamps: there are 11 + 20 + 7 = 38 stamps in total.

Conclusion: Bella bought 38 stamps in all. |

Table 11: Step-Back few-shot demonstration exemplar for GSM8K.

| Knowledge QA Step-Back Prompt |
| --- |
| You are an expert at world knowledge. Your task is to step back and paraphrase a question to a more generic step-back question, which is easier to answer. Here are a few examples:

Original Question: <Original Question Example1>
Stepback Question: <Stepback Question Example1>
...
Original Question: <Original Question Example5>
Stepback Question: <Stepback Question Example5>
Original Question: <Original Question>
Stepback Question: |

Table 12: Prompt of asking step-back question in Knowledge QA tasks.

| dataset | Original Question | Step-back Question |
| --- | --- | --- |
| TimeQA | Which position did Knox Cunningham hold from May 1955 to Apr 1956? | Which positions have Knox Cunningham held in his career? |
| TimeQA | Who was the spouse of Anna Karina from 1968 to 1974? | Who were the spouses of Anna Karina? |
| TimeQA | Which team did Thierry Audel play for from 2007 to 2008? | Which teams did Thierry Audel play for in his career? |
| TimeQA | What was the operator of GCR Class 11E from 1913 to Dec 1922? | What were the operators of GCR Class 11E in history? |
| TimeQA | Which country did Sokolovsko belong to from 1392 to 1525? | Which countries did Sokolovsko belong to in history? |
| SituatedQA | when was the last time a team from canada won the stanley cup as of 2002 | which years did a team from canada won the stanley cup as of 2002 |
| SituatedQA | when did england last get to the semi final in a world cup as of 2019 | which years did england get to the semi final in a world cup as of 2019? |
| SituatedQA | what is the biggest hotel in las vegas nv as of November 28, 1993 | what is the size of the hotels in las vegas nv as of November 28, 1993 |
| SituatedQA | who has scored most runs in t20 matches as of 2017 | What are the runs of players in t20 matches as of 2017 |
| SituatedQA | who is the highest paid player in the nba this season as of 2017 | what is the salary of the high paid players in the nba this season as of 2017 |

Table 13: Few-shot demonstration exemplars for asking step-back questions in TimeQA and SituatedQA.

| Knowledge QA Final-Answer Prompt |
| --- |
| You are an expert of world knowledge. I am going to ask you a question. Your response should be comprehensive and not contradicted with the following context if they are relevant. Otherwise, ignore them if they are not relevant.

<Passage from original retrieval augmentation>
<Passage from step-back retrieval augmentation>

Original Question: <Original Question>
Answer: |

Table 14: Prompt of querying the model for final answer with additional contexts from original and step-back retrieval augmentations in TimeQA and SituatedQA

| dataset | Original Question | Step-back Question |
| --- | --- | --- |
| MuSiQue | at year saw the creation of the region where the county of Hertfordshire is located? | which region is the county of Hertfordshire located? |
| MuSiQue | Jan Šindel's was born in what country? | what is Jan Šindel's personal history? |
| MuSiQue | When was the abolishment of the studio that distributed The Game? | which studio distributed The Game? |
| MuSiQue | What city is the person who broadened the doctrine of philosophy of language from? | who broadened the doctrine of philosophy of language |
| MuSiQue | When was the baseball team winning the world series in 2015 baseball created? | which baseball team won the world series in 2015 baseball? |
| StrategyQA | Could the members of The Police perform lawful arrests? | what can the members of The Police do? |
| StrategyQA | Would a Monoamine Oxidase candy bar cheer up a depressed friend? | What are the effects of Monoamine Oxidase? |
| StrategyQA | Would a dog respond to bell before Grey seal? | Would a dog respond to bell before Grey seal? |
| StrategyQA | Is shrimp scampi definitely free of plastic? | what is shrimp scampi made of? |
| StrategyQA | Do the anchors on Rede Globo speak Chinese? | What languages do the anchors on Rede Globo speak? |

Table 15: Few-shot demonstration exemplars for asking step-back questions in MuSiQue and StrategyQA.

| Baseline few-shot Prompt |
| --- |
| You are an expert of world knowledge and physics. Your task is to solve the following question. Here are a few examples:

Question: \<Question Example>
Answer: \<Answer Example>
Question: \<Question>
Answer: |

Table 16: Prompt of querying the baseline model for final answer with few-shot demonstration exemplars.

For few-shot CoT prompting, we use the same template as the Baseline prompting in Sec. D.4 by replacing the few-shot examples using CoT responses, as shown in Tables 18, 19, 20, 21, and 22.

### D.6 TAKE A DEEP BREATH (TDB)

We study the zero-shot prompting found in Yang et al. (2023): we take *Take a deep breath and work on this problem step-by-step*, and prepend it to the question.

## E EXAMPLES OF ERROR ANALYSIS AND WINS OF STEP-BACK PROMPTING

### E.1 MMLU ERROR ANALYSIS

In Tables 23-27, we show one example for each of the 5 error categories we identified through error analysis on STEP-BACK PROMPTING.

### E.2 EXAMPLE WINS FROM STEP-BACK PROMPTING

Tables 28- 30, 31 32, 33 and 34 illustrate the some successful examples of STEP-BACK PROMPTING on MMLU-Physics, MMLU-Chemistry, TimeQA, SituatedQA, and StrategyQA respectively.

| dataset | Question | Answer |
|---|---|---|
| MMLU Physics | A point charge, Q1 = +1 mC, is fixed at the origin. How much work is required to move another charge, Q2 = +8 $\mu$C, from the point (0, 4 meters) to the point (3 meters, 0)?
0) 3.5 J 1) 6.0 J 2) 22.5 J 3) 40 J | The work required to move a charge in an electric field is equal to the change in potential energy of the charge. The potential energy of a charge in an electric field is given by: U = k * Q1 * Q2 / r where: k is the Coulomb constant, Q1 and Q2 are the charges, r is the distance between the charges. In this case, the initial potential energy of Q2 is: U1 = k * Q1 * Q2 / 4 meters

The final potential energy of Q2 is: U2 = k * Q1 * Q2 / 3 meters

The change in potential energy is: $\Delta$U = U2 - U1 = k * Q1 * Q2 * (1/3 meters - 1/4 meters)

Plugging in the values, we get: $\Delta U = (9 \times 10^9 N * m^2/C^2) \times (1 \times 10^{-3}C) * (8 \times 10^{-6}C) * (1/3m - 1/4m) = 6J$. The answer is 1) 6.0 J. |
| MMLU Chemistry | A sample of an unknown chloride compound was dissolved in water, and then titrated with excess Pb(NO3)2 to create a precipitate. After drying, it is determined there are 0.0050 mol of precipitate present. What mass of chloride is present in the original sample?
0) 0.177 g 1) 0.355 g 2) 0.522 g 3) 0.710 g | 0.0050 moles of lead chloride precipitate contain 0.010 moles of chloride ions due to the 2:1 mole ratio between chloride ions and lead chloride.

The mass of 0.010 moles of chloride ions is 0.355 g.

Therefore, the mass of chloride present in the original sample is 0.355 g. |
| GSM8K | Bella bought stamps at the post office. Some of the stamps had a snowflake design, some had a truck design, and some had a rose design. Bella bought 11 snowflake stamps. She bought 9 more truck stamps than snowflake stamps, and 13 fewer rose stamps than truck stamps. How many stamps did Bella buy in all? | The number of truck stamps is 11 + 9 = 20. The number of rose stamps is 20 - 13 = 7. Bella bought 11 + 20 + 7 = 38 stamps in all. |
| TimeQA | Who was the spouse of Anna Karina from 1968 to 1974? | Anna Karina's spouse from 1968 to 1974 was French actor Pierre Fabre. |
| SituatedQA | what is the biggest hotel in las vegas nv as of November 28, 1993 | The biggest hotel in Las Vegas, Nevada as of November 28, 1993 was the Excalibur Hotel & Casino. |
| MuSiQue | What year saw the creation of the region where the county of Hertfordshire is located? | 1994 |
| StrategyQA | Would a Monoamine Oxidase candy bar cheer up a depressed friend? | A Monoamine Oxidase (MAO) candy bar would not cheer up a depressed friend. MAO is an enzyme that breaks down neurotransmitters such as serotonin, dopamine, and norepinephrine. These neurotransmitters are associated with mood, so inhibiting MAO can lead to an increase in their levels. |

Table 17: Demonstration exemplar for standard prompting.

| dataset | Question | Answer |
|---|---|---|
| MMLU Physics | A point charge, Q1 = +1 mC, is fixed at the origin. How much work is required to move another charge, Q2 = +8 $\mu$C, from the point (0, 4 meters) to the point (3 meters, 0)?
0) 3.5 J
1) 6.0 J
2) 22.5 J
3) 40 J | Sure, let's think step by step.

1. Calculate the electric potential energy of Q2 at the initial position. The electric potential energy (U) of a point charge Q2 at a distance r from another point charge Q1 is given by the following equation: U = k * Q1 * Q2 / r, where k is Coulomb's constant.

Therefore, the electric potential energy of Q2 at the initial position is:
$U_{\text{initial}} = (9 \times 10^9 N m^2/C^2) * (1 \times 10^{-3}C) * (8 \times 10^{-6}C)/4m$ = 18 J

2. Calculate the electric potential energy of Q2 at the final position. The electric potential energy of Q2 at the final position is:
$U_{\text{final}} = (9 \times 10^9 N m^2/C^2) * (1 \times 10^{-3}C) * (8 \times 10^{-6}C)/3m$ = 24 J

3. Calculate the work required to move Q2 from the initial position to the final position. The work (W) required to move a charge from one position to another is equal to the change in electric potential energy ($\Delta$U) between the two positions.
W = $\Delta$U = $U_{\text{final}}$ - $U_{\text{initial}}$ = 24 J - 18 J = 6 J

Therefore, the answer is 1) 6.0 J. |

Table 18: Demonstration exemplar for CoT prompting for MMLU Physics.

| dataset | Question | Answer |
|---------|----------|--------|
| MMLU Chemistry | A sample of an unknown chloride compound was dissolved in water, and then titrated with excess Pb(NO3)2 to create a precipitate. After drying, it is determined there are 0.0050 mol of precipitate present. What mass of chloride is present in the original sample?
0) 0.177 g
1) 0.355 g
2) 0.522 g
3) 0.710 g | Here's the step-by-step solution:
Identify the precipitate: The precipitate formed when a chloride compound reacts with lead nitrate (Pb(NO3)2) is lead chloride (PbCl2).

Determine the mole ratio: The balanced chemical equation for the reaction between a chloride compound (represented as MCl, where M is the unknown metal) and lead nitrate is: $2MCl + Pb(NO_3)_2 \longrightarrow 2MNO_3 + PbCl_2$
From the balanced equation, you can see that 2 moles of chloride ions (Cl-) react with 1 mole of lead nitrate to produce 1 mole of lead chloride precipitate. Additionally, 1 mole of lead chloride contains 2 moles of chloride ions.

Calculate the moles of chloride ions: Since 0.0050 moles of lead chloride precipitate were formed, there must have been 2 × 0.0050 = 0.010 moles of chloride ions in the original sample.

Calculate the mass of chloride ions: The molar mass of chlorine (Cl) is 35.45 g/mol. Therefore, the mass of 0.010 moles of chloride ions is:
mass = moles × molar mass = 0.010 mol × 35.45 g/mol = 0.355 g

Therefore, the mass of chloride present in the original sample is 0.355 g. The correct answer is option 1. |

Table 19: Demonstration exemplar for CoT prompting for MMLU Chemistry.

| dataset | Question | Answer |
|---------|----------|--------|
| GSM8K | Bella bought stamps at the post office. Some of the stamps had a snowflake design, some had a truck design, and some had a rose design. Bella bought 11 snowflake stamps. She bought 9 more truck stamps than snowflake stamps, and 13 fewer rose stamps than truck stamps. How many stamps did Bella buy in all? | Here are the steps to solve the problem:

Step 1: Find the number of truck stamps. Adding 9 to the number of snowflake stamps gives the number of truck stamps: 11 + 9 = 20

Step 2: Find the number of rose stamps. Subtracting 13 from the number of truck stamps gives the number of rose stamps: 20 - 13 = 7

Step 3: Add the number of snowflake, truck, and rose stamps to find the total number of stamps Bella bought: 11 + 20 + 7 = 38

Solution: Bella bought 38 stamps in all. |

Table 20: Demonstration exemplar for CoT prompting for GSM8K.

| dataset | Question | Answer |
|---------|----------|--------|
| TimeQA | Who was the spouse of Anna Karina from 1968 to 1974? | Step 1: Identify Anna Karina's spouses:Jean-Luc Godard (1961–1965). Pierre Fabre (1968–1974) Daniel Duval (1978–1981) Dennis Berry (1982–1994)  Step 2: Determine which spouse was married to Anna Karina from 1968 to 1974. Jean-Luc Godard was married to Anna Karina from 1961 to 1965, so he is not the correct answer. Pierre Fabre was married to Anna Karina from 1968 to 1974, so he is the correct answer. Daniel Duval was married to Anna Karina from 1978 to 1981, so he is not the correct answer. Dennis Berry was married to Anna Karina from 1982 to 1994, so he is not the correct answer.  Therefore, Anna Karina's spouse from 1968 to 1974 was Pierre Fabre. |
| SituatedQA | what is the biggest hotel in las vegas nv as of November 28, 1993 | Sure, let's think step by step: What were the largest hotels in Las Vegas in 1993? According to a 1993 article in the Las Vegas Review-Journal, the largest hotels in Las Vegas at the time were: MGM Grand (5,005 rooms) Excalibur (4,008 rooms) The Mirage (3,049 rooms) Flamingo Hilton (3,565 rooms) Caesars Palace (2,577 rooms) Which of these hotels was the largest on November 28, 1993? The MGM Grand opened in December 1993, so it would not have been the largest hotel on November 28, 1993. The Excalibur opened in June 1990, so it would have been the largest hotel on November 28, 1993. Therefore, the largest hotel in Las Vegas on November 28, 1993 was the Excalibur. |

Table 21: Demonstration exemplar for CoT prompting for TimeQA and SituatedQA.

| dataset | Question | Answer |
|---------|----------|--------|
| MuSiQue | What year saw the creation of the region where the county of Hertfordshire is located? | The county of Hertfordshire is in the 'East of England' region. The 'East of England' region was created in 1994. So, the answer is 1994. |
| StrategyQA | Would a Monoamine Oxidase candy bar cheer up a depressed friend? | No, a Monoamine Oxidase (MAO) candy bar would not cheer up a depressed friend. In fact, it could be dangerous for them to consume. Here's a step-by-step explanation:
1. MAO is an enzyme that breaks down neurotransmitters like serotonin, dopamine, and norepinephrine. These neurotransmitters are important for regulating mood, and low levels of them are associated with depression.
2. MAO inhibitors (MAOIs) are a type of medication that blocks the action of MAO, which can increase levels of these neurotransmitters in the brain. MAOIs are sometimes used to treat depression, but they can have serious side effects, including interactions with certain foods and medications.
3. A MAO candy bar would not contain enough MAOI to have a therapeutic effect on depression. In fact, it is unlikely that a candy bar could contain any MAOI at all, as these medications are typically taken in pill form.
In conclusion, a MAO candy bar would not cheer up a depressed friend and could even be dangerous for them to consume. |

Table 22: Demonstration exemplar for CoT prompting for MuSiQue and StrategyQA.

| Question | An alarm whose frequency is 400 Hz is dropped out of a third-floor window. The student who drops it measures the frequency with a very sensitive oscilloscope. The measured frequency
0) appears higher than 400 Hz and the frequency increase as it falls
1) appears higher than 400 Hz and the frequency decreases as it falls
2) appears lower than 400 Hz and the frequency decreases as it falls
3) appears lower than 400 Hz and the frequency increases as it falls |
|---|---|
| Principles | Doppler Effect: the effect produced by the motion of a wave source, observer, or both that causes a change in wave frequency perceived by an observer relative to the frequency produced by the source. The Doppler effect is observed with all types of waves, including electromagnetic and acoustic waves. It was named after Austrian physicist Christian Doppler. |
| Answer | Using the Principles of Doppler Effect, we can solve the problem as following:

Step 1: Apply Doppler Effect to find out the measured frequency.
The Doppler Effect is the change in frequency of a wave as measured by an observer who is moving relative to the source of the wave. In this case, the source of the wave is the alarm, and the observer is the student who drops it.

As the alarm falls, the distance between the alarm and the student decreases. This means that the frequency of the wave as measured by the student will increase. Therefore, the measured frequency will appear higher than 400 Hz, and the frequency will increase as the alarm falls.

Therefore, the correct answer is 0) appears higher than 400 Hz and the frequency increase as it falls. |

Table 23: Error example of STEP-BACK PROMPTING on MMLU high-school Physics in the class of **Factual Error**. In this case, the Factual error leads to the wrong answer despite that the Principle and Reasoning are all sound.

| Question | An astronaut lands on a planet whose mass and radius are each twice that of Earth. If the astronaut weighs 800 N on Earth, how much will he weigh on this planet?
0) 200 N
1) 400 N
2) 800 N
3) 1600 N |
|---|---|
| Principles | Weight: W = m * g, where W is the weight, m is the mass and g is the acceleration due to gravity
Acceleration due to gravity: $g = G * m / r^2$, where G is the gravitational constant, m is the mass of the object and r is the radius of the object |
| Answer | Using the Principles of Newton's law of universal gravitation and Weight, we can solve the problem as following:

Step 1: Apply Newton's law of universal gravitation to find out the acceleration due to gravity on the planet.
The acceleration due to gravity on a planet is given by the formula:
$g = G * m / r^2$, where g is the acceleration due to gravity, G is the gravitational constant, m is the mass of the planet, and r is the radius of the planet.

In this problem, we are given the following information:
$m = 2 * m_{earth}$
$r = 2 * r_{earth}$

Plugging these values into the formula, we get:
$g = G * 2 * m_{earth} / (2 * r_{earth})^2$
$g = G * m_{earth} / r_{earth}^2$

This is the same as the acceleration due to gravity on Earth.

Step 2: Apply Weight to find out the weight of the astronaut on the planet.
The weight of an object is given by the formula:
W = m * g, where W is the weight, m is the mass of the object, and g is the acceleration due to gravity.

In this problem, we are given the following information:
$m = 800 \text{ N} / g_{earth}$
$g = g_{earth}$

Plugging these values into the formula, we get:
$W = 800 \text{ N} / g_{earth} * g_{earth}$
$W = 800 \text{ N}$

Therefore, the astronaut will weigh 800 N on the planet. The correct answer is 2) 800 N. |

Table 24: Error example of STEP-BACK PROMPTING on MMLU high-school Physics in the class of **Math Error**. As highlighted, the math error leads to directly a wrong answer with a factor of 2 off.

| Question | Two spheres of net charge +5e and -6e briefly come into contact. Afterward, which of the following is a possible combination of net charges for the two spheres? 
 0) +3e and -4e 
 1) +1.5e and -2.5e 
 2) Both of the above 
 3) None of the above |
|---|---|
| Principles | Law of conservation of charge: the total charge in a closed system remains constant. 
 Coulomb's Law: the mutual repulsion of like charges demands that the charges be as far apart as possible. |
| Answer | Using the Principles of Conservation of charge, we can solve the problem as following: 

 Step 1: Apply Conservation of charge to find out the possible combination of net charges for the two spheres. 
 The total charge of the two spheres before they come into contact is +5e - 6e = -1e. After they come into contact, the total charge of the two spheres must still be -1e. 

 Step 2: Find out the possible combination of net charges for the two spheres. 
 The possible combination of net charges for the two spheres are: 
 (1) +3e and -4e 
 (2) +1.5e and -2.5e 

 Therefore, the correct answer is 2) Both of the above. |

Table 25: Error example of STEP-BACK PROMPTING on MMLU high-school Physics in the class of **Principle Error**. In this case, Coulomb's Law is not needed for solving this problem. Instead, the Charge Quantization Principle is missing from the retrieved principles, and is in fact needed to rule out option (2) of fractional charges.

| | |
|---|---|
| Question | The circumference of a helium-filled balloon is measured for three different conditions: at room temperature, after being in a warm oven for 30 minutes, and after being in a freezer for 30 minutes. A student plotting the circumference cubed $C^3$ as a function of temperature T, should expect to find which of the following?
0) a cubic relationship between $C^3$ and T
1) an indirect relationship between $C^3$ and T
2) an extrapolated temperature T where $C^3$ reaches zero
3) a maximum $C^3$ as the temperature T increases |
| Principles | Ideal Gas Law: PV = nRT, where P is the pressure, V is the volume, n is the number of moles of gas, R is the ideal gas constant, T is the absolute temperature.
Volume of a sphere: $V = 4/3 * \pi * r^3$, where r is the radius.
Absolute temperature: Absolute temperature is always written as kelvin.
Absolute zero: Absolute zero is the lowest limit of the kelvin scale and is the lowest possible temperature (- 273.15 °C). |
| Answer | Using the Principles of Charles's Law and Gay-Lussac's Law, we can solve the problem as following:
Step 1: Apply Charles's Law to find out the relationship between the volume and temperature of the gas. Charles's Law states that the volume of a fixed mass of a gas is directly proportional to the absolute temperature of the gas when pressure is kept constant. This means that if the temperature of the gas increases, the volume of the gas will also increase.

Step 2: Apply Gay-Lussac's Law to find out the relationship between the pressure and temperature of the gas. Gay-Lussac's Law states that the pressure of a fixed mass of a gas is directly proportional to the absolute temperature of the gas when volume is kept constant. This means that if the temperature of the gas increases, the pressure of the gas will also increase.

Step 3: Apply Charles's Law and Gay-Lussac's Law to find out the relationship between the circumference and temperature of the gas.The circumference of the balloon is directly proportional to the volume of the gas. This is because the circumference of the balloon is the distance around the outside of the balloon, and the volume of the gas is the amount of space that the gas takes up.

The pressure of the gas is directly proportional to the temperature of the gas. This is because the pressure of the gas is the force that the gas exerts on the walls of the balloon, and the temperature of the gas is the measure of the average kinetic energy of the gas molecules.

Therefore, the circumference of the balloon is directly proportional to the temperature of the gas. This means that if the temperature of the gas increases, the circumference of the balloon will also increase.

Step 4: Apply the relationship between the circumference and temperature of the gas to find out the relationship between the circumference cubed and temperature of the gas. The circumference cubed is directly proportional to the temperature of the gas. This is because the circumference cubed is the volume of the balloon, and the volume of the balloon is directly proportional to the temperature of the gas.

Therefore, the student should expect to find a cubic relationship between $C^3$ and T. The correct answer is 0) a cubic relationship between $C^3$ and T. |

Table 26: Error example of STEP-BACK PROMPTING on MMLU high-school Physics in the class of **Context Loss**. In this case, at the end of the Reasoning chain, the model forgot the original question, and lose the context to apply the reasoning to the question.

| Question | Physics students are checking the constant acceleration equations of kinematics by measuring the velocity of a tennis ball that is dropped and falls 6 meters and then passes through a photogate. The predicted velocity is $20\%$ above the velocity measured by the photogate. Which of the following best describes the cause of the large percent difference?
0) The ball changes its shape while falling.
1) The acceleration of gravity varies as the ball is falling.
2) Air resistance increases the acceleration of the ball.
3) The acceleration of the balls varies with the velocity. |
|---|---|
| Principles | Vertical motion: y = 0.5 * g * $t^2$, where y is the vertical displacement, g is the acceleration due to gravity, t is the time.
Free fall: is a special case of motion with constant acceleration, in which the acceleration is due to gravity.
Acceleration to Velocity Formula: v = u + a * t, where v is the final speed, u is the initial velocity, a is the acceleration, and t is the time. |
| Answer | Using the Principles of Vertical motion and Horizontal motion, we can solve the problem as following:

Step 1: Apply the Vertical motion equation to find the time it takes for the ball to fall 6 meters.
y = 0.5 * g * $t^2$
6 m = 0.5 * 9.8 m/$s^2$ * $t^2$
t = 1.12 s

Step 2: Apply the Horizontal motion equation to find the velocity of the ball when it passes through the photogate.
x = v * t
v = x / t
v = 6 m / 1.12 s
v = 5.36 m/s

Step 3: Compare the predicted velocity with the velocity measured by the photogate. The predicted velocity is $20\%$ above the velocity measured by the photogate, so the predicted velocity is 1.2 * 5.36 m/s = 6.43 m/s.
Step 4: Find the cause of the large percent difference. The predicted velocity is 6.43 m/s, but the velocity measured by the photogate is 5.36 m/s. The difference is 1.07 m/s. This difference is caused by air resistance. Air resistance is a force that opposes the motion of an object through the air. It is caused by the friction between the object and the air molecules. The greater the surface area of the object, the greater the air resistance. The tennis ball has a large surface area, so it experiences a lot of air resistance. This air resistance slows down the tennis ball and causes the difference between the predicted velocity and the velocity measured by the photogate.
Therefore, the correct answer is 0) The ball changes its shape while falling. |

Table 27: Error example of STEP-BACK PROMPTING on MMLU high-school Physics in the class of **Reasoning Error**. In this case, the model was able to correctly identify air resistance is the cause of the velocity difference, but made the error at the final reasoning step of deriving the right answer.

| dataset | example |
|---------|---------|
| MMLU-Physics | **Original Question**: What happens to the pressure, P, of an ideal gas if the temperature is increased by a factor of 2 and the volume is increased by a factor of 8 ?
0) P decreases by a factor of 16.
1) P decreases by a factor of 4.
2) P decreases by a factor of 2.
3) P increases by a factor of 4.
**Original Answer [Incorrect]**: The correct answer is 0. The ideal gas law is PV = nRT. If the temperature is increased by a factor of 2 and the volume is increased by a factor of 8, then the pressure will decrease by a factor of 16.
**Stepback Question**: What are the physics principles behind this question?
**Principles**:
Ideal gas law: PV = nRT, where P is the pressure, V is the volume, n is the number of moles, R is the gas constant, and T is the temperature.
**Final Answer**: Using the Principles of Ideal gas law, we can solve the problem as following:
Step 1: Apply Ideal gas law to find out the pressure of an ideal gas. The ideal gas law states that PV = nRT, where P is the pressure, V is the volume, n is the number of moles, R is the gas constant, and T is the temperature.

Step 2: Apply Ideal gas law to find out the pressure of an ideal gas when the temperature is increased by a factor of 2 and the volume is increased by a factor of 8. If the temperature is increased by a factor of 2, then T becomes 2T. If the volume is increased by a factor of 8, then V becomes 8V.

Substituting these values into the ideal gas law, we get: P(8V) = nR(2T)

Dividing both sides by 8V, we get: P = nR(2T) / 8V

We can see that the pressure has decreased by a factor of 4.

Therefore, the correct answer is 1) P decreases by a factor of 4. |

Table 28: Illustration of wins of STEP-BACK PROMPTING on the MMLU-Physics dataset.

| dataset | example |
|---------|---------|
| MMLU-Physics | **Original Question**: A liquid flows at a constant flow rate through a pipe with circular cross-sections of varying diameters. At one point in the pipe, the diameter is 2 cm and the flow speed is 18 m/s. What is the flow speed at another point in this pipe, where the diameter is 3 cm?
0) 4 m/s
1) 6 m/s
2) 8 m/s
3) 12 m/s
**Original Answer [Incorrect]**: The answer is 3) 12 m/s A step-by-step solution is below: The flow rate is constant, so the product of the speed and the cross-sectional area is constant. The cross-sectional area is proportional to the square of the diameter, so the speed is inversely proportional to the square of the diameter. The ratio of the speeds is the inverse of the ratio of the squares of the diameters, so the speed at the second point is $18m/s * (2cm/3cm)^2 = 12$ m/s.
**Stepback Question**: What are the physics principles behind this question?
**Principles**:
Conservation of Mass: the amount of mass flowing through the pipe at two different points must be equal, or A1 * v1 = A2 * v2, where A1 is the area of cross-section one, v1 is the velocity at point 1, A2 is the area of cross-section two, v2 is the velocity at point 2.
Area of Circle: $A = \pi * r^2$, where r is the radius of the circle.
**Final Answer**: Using the Principles of Continuity equation, we can solve the problem as following:
Step 1: Apply the Continuity equation to find out the flow speed at another point in this pipe. The continuity equation states that the product of the area and the velocity is constant at all points in a pipe. In this problem, we are given the following information:
$A1 = \pi * (2cm)^2 = 4\pi cm^2$
v1 = 18 m/s
$A2 = \pi * (3cm)^2 = 9\pi cm^2$

We can use the continuity equation to find v2:
A1 * v1 = A2 * v2
$4\pi cm^2 * 18m/s = 9\pi cm^2 * v2$
v2 = 8 m/s

Therefore, the flow speed at another point in this pipe is 8 m/s. The correct answer is 2) 8 m/s. |

Table 29: Illustration of wins of STEP-BACK PROMPTING on the MMLU-Physics dataset.

| dataset | example |
|---------|---------|
| MMLU-Physics | **Original Question**: A 2 $\mu$F capacitor is connected directly to a battery. When the capacitor is fully charged, it stores 600 $\mu$C of charge. An experimenter replaces the 2 $\mu$F capacitor with three 18 $\mu$F capacitors in series connected to the same battery. Once the capacitors are fully charged, what charge is stored on each capacitor?
0) 100 $\mu$C
1) 200 $\mu$C
2) 600 $\mu$C
3) 1800 $\mu$C
**Original Answer [Incorrect]**: The answer is 100 $\mu$C. The total capacitance of the three 18 $\mu$F capacitors in series is 1 / (1 / 18 + 1 / 18 + 1 / 18) = 6 $\mu$F. The charge stored on the three capacitors in series is 6 $\mu$F * V = 600 $\mu$C. The charge stored on each capacitor is 600 $\mu$C / 3 = 200 $\mu$C.
**Stepback Question**: What are the physics principles behind this question?
**Principles**:
Capacitors: Q = C *V, where Q is the charge, C is the capacitance and V is the voltage.
Capacitors in series: C = 1 / (1/C1 + 1/C2 + ... + 1/Cn), where C is the equivalent capacitance and C1, C2, Cn are the capacitances of each capacitor.
Capacitor in parallel: C = C1 + C2 + ... + Cn, where C is the equivalent capacitance and C1, C2, Cn are the capacitances of each capacitor.
**Final Answer**: Using the Principles of Capacitance and Capacitors in series, we can solve the problem as following:
Step 1: Apply Capacitance to find out the voltage of the battery.
We know that the capacitance of the 2 $\mu$F capacitor is 2 $\mu$F and the charge stored on the capacitor is 600 $\mu$C. Using the formula for capacitance, we can find the voltage of the battery:
C = Q / V
2 $\mu$F = 600 $\mu$C / V
V = 300 V

Step 2: Apply Capacitors in series to find out the equivalent capacitance of the three 18 $\mu$F capacitors in series.
We know that the capacitance of each of the three capacitors is 18 $\mu$F. Using the formula for capacitors in series, we can find the equivalent capacitance of the three capacitors:
$1 / C_{eq} = 1 / C_1 + 1 / C_2 + 1 / C_3$
$1 / C_{eq} = 1 / 18 \ \mu$F$ + 1 / 18 \ \mu$F$ + 1 / 18 \ \mu$F
$1 / C_{eq} = 3 / 18 \ \mu$F
$C_{eq} = 6 \ \mu$F

Step 3: Apply Capacitance to find out the charge stored on each capacitor.
We know that the equivalent capacitance of the three capacitors is 6 $\mu$F and the voltage of the battery is 300 V. Using the formula for capacitance, we can find the charge stored on each capacitor:
C = Q / V
6 $\mu$F = Q / 300 V
Q = 1800 $\mu$C

Therefore, the charge stored on each capacitor is 1800 $\mu$C. The correct answer is 3) 1800 $\mu$C. |

Table 30: Illustration of wins of STEP-BACK PROMPTING on the MMLU-Physics dataset.

| dataset | example |
|---|---|
| MMLU-Chemistry | **Original Question**: Which of the following indicates that a reaction is spontaneous?
0) at equilibrium there are more products than reactants
1) the value of $\Delta G^0$ is greater than zero
2) the value of $\Delta S^0$ is greater than zero
3) the value of Keq is less than one
**Original Answer [Incorrect]**: 2) the value of $\Delta S°$ is greater than zero.
A reaction is spontaneous if the value of $\Delta S°$ is greater than zero. This is because a positive $\Delta S°$ indicates that the reaction results in an increase in entropy, which is a measure of disorder. A spontaneous reaction is one that occurs without the input of energy, and an increase in entropy is a natural tendency of the universe.
**Stepback Question**: What are the chemistry principles behind this question?
**Principles**:
Spontaneous reactions: A spontaneous reaction is a reaction that occurs without the input of energy.
**Final Answer**: A reaction is spontaneous if the change in Gibbs free energy is negative ($\Delta G < 0$). The value of $\Delta G^0$ is greater than zero indicates that the reaction is not spontaneous.

The correct answer is 0) at equilibrium there are more products than reactants. |
| MMLU-Chemistry | **Original Question**: Potassium-40 is a minor isotope found in naturally occurring potassium. It is radioactive and can be detected on simple radiation counters. How many protons, neutrons, and electrons does potassium-40 have when it is part of K2SO4?
0) 21 neutrons, 19 protons, 18 electrons
1) 20 neutrons, 19 protons, 19 electrons
2) 21 neutrons, 19 protons, 19 electrons
3) 19 neutrons, 19 protons, 19 electrons
**Original Answer [Incorrect]**: The correct answer is 2).
Potassium-40 has 19 protons and 21 neutrons. When it is part of K2SO4, it has 19 electrons.
**Stepback Question**: What are the chemistry principles behind this question?
**Principles**:
Atomic number: The atomic number of an element is the number of protons in the nucleus of an atom of that element.
**Final Answer**: The atomic number of potassium is 19, so all potassium atoms have 19 protons. The mass number of potassium-40 is 40, so it has 40 - 19 = 21 neutrons. When potassium is part of a compound, it loses one electron to become a potassium ion. Therefore, potassium-40 has 19 - 1 = 18 electrons when it is part of K2SO4.
The correct answer is 0) 21 neutrons, 19 protons, 18 electrons. |

Table 31: Illustration of wins of STEP-BACK PROMPTING on the MMLU-Chemistry dataset.

| dataset | example |
|---------|---------|
| TimeQA | **Original Question**: Carlos Ascues played for which team from 2013 to 2014?
**Original Answer [Incorrect]**: Carlos Ascues played for Benfica from 2013 to 2014.
**Stepback Question**: Which teams did Carlos Ascues play for?
**Final Answer [Correct]**: From 2013 to 2014, Carlos Ascues played for the Greek team Panetolikos. |
| TimeQA | **Original Question**: Seth Nana Twumasi played for which team between Jan 2011 and Feb 2011?
**Original Answer [Incorrect]**: Seth Nana Twumasi played for the FC Honka team between Jan 2011 and Feb 2011.
**Stepback Question**: Which teams did Seth Nana Twumasi play for?
**Final Answer [Correct]**: Northampton. He was on a loan transfer from Peterborough. |
| TimeQA | **Original Question**: What position did Carmel Hanna take in May 2002?
**Original Answer [Incorrect]**: According to the article, Carmel Hanna was appointed as Minister of State for Children and the Family in May 2002. She held this position until 2005.
**Stepback Question**: What positions did Carmel Hanna hold?
**Final Answer [Correct]**: From the context, we can see that Carmel Hanna was named minister of Employment and Learning (2001). We can then infer that she held the position of Minister for Employment and Learning in May 2002. |

Table 32: Illustration of wins of STEP-BACK PROMPTING on the TimeQA dataset.

| dataset | example |
|---------|---------|
| SituatedQA | **Original Question**: when was the last time miss texas won miss america as of November 29, 2010
**Original Answer [Incorrect]**: Laura Leigh Covington was the last Miss Texas to win the Miss America competition. This happened in 2000.
**Stepback Question**: which years did miss texas win miss america
**Final Answer [Correct]**: According to Wikipedia, the last time Miss Texas won Miss America was in 1975. |
| SituatedQA | **Original Question**: when is the last time congress passed a complete budget as of 2021
**Original Answer [Incorrect]**: The last time Congress passed a complete budget as of 2021 was on January 19, 2018.
**Stepback Question**: what are the years that congress passed a complete budget as of 2021
**Final Answer [Correct]**: The last time Congress passed a complete budget as of 2021 was December 27, 2020, the Consolidated Appropriations Act, 2021. |
| SituatedQA | **Original Question**: when was the last time unc did not make the ncaa tournament as of 2003
**Original Answer [Incorrect]**: The last time UNC did not make the NCAA tournament was 1990.
**Stepback Question**: which years did unc not make the ncaa tournament as of 2003
**Final Answer [Correct]**: The correct answer is 2002-03. The context shows the standings of North Carolina Tar Heels men's basketball. From the standings, we can see that this team has 19 wins and 16 losses. 19-16 is not good enough to make the NCAA tournament, which is mentioned in the context. From the standings, we can also see that this is the standing for 2002-03 season. So the last time UNC did not make the NCAA tournament as of 2003 is 2002-03. |

Table 33: Illustration of wins of STEP-BACK PROMPTING on the SituatedQA dataset.

| dataset | example |
|---------|---------|
| StrategyQA | **Original Question**: Could you drive a Rowe 550 to the 2008 Summer Olympics?
**Original Answer [Incorrect]**: The 2008 Summer Olympics was held in Beijing, China. The Rowe 550 is an all-electric car. The Rowe 550 has a range of 70 miles on a single charge and a top speed of about 30 mph. The distance between Beijing and the U.K. is about 4000 miles.
**Stepback Question**: What kind of vehicle is a Rowe 550?
**Final Answer [Correct]**: The Roewe 550 was released in 2008; the 2008 Summer Olympics were held from August 8-24, 2008. So you could drive a Rowe 550 to the 2008 Summer Olympics. |
| StrategyQA | **Original Question**: what channel does thursday night football come on regular tv as of 2017?
**Original Answer [Incorrect]**: Thursday Night Football is a weekly series of National Football League (NFL) games broadcast on Thursday nights. It started in 2006 on NFL Network. Under a new agreement between the NFL and CBS, CBS will air 50% of the Thursday Night Football games and NFL Network will air the other 50%.
**Stepback Question**: which channel is thursday night football on as of 2017?
**Final Answer [Correct]**: In 2017, Thursday Night Football was aired on CBS, NBC, and NFL Network. |
| StrategyQA | **Original Question**: who is the present president of assam sahitya sabha as of 2018?
**Original Answer [Incorrect]**: Dr. Nagen Saikia
**Stepback Question**: who lead assam sahitya sabha as of 2018?
**Final Answer [Correct]**: The answer is Paramananda Rajbongshi. As per a news from 2019, The Asam Sahitya Sabha, which was established in 1917, in Sibsagar, Assam, had got Paramananda Rajbongshi as its president. So, as of 2018, the president of assam sahitya sabha is Paramananda Rajbongshi. |

Table 34: Illustration of wins of STEP-BACK PROMPTING on the StrategyQA dataset.

