# OpenReview forum: "Take a Step Back: Evoking Reasoning via Abstraction in Large Language Models"
_ICLR.cc/2024/Conference — ICLR 2024 poster_

### Official Review · Reviewer_XiDg · 2023-10-30

**Soundness:** 3 good
**Presentation:** 3 good
**Contribution:** 3 good
**Rating:** 8
**Confidence:** 4

**Summary:**

The authors introduce a new form of guided prompting for Q&A settings in which a model 1) *Abstracts* (takes a step back) the key concepts relevant to answering a given question and then 2) *Reasons* by using the Step-back answer in conjunction with the original question to produce a final answer. The authors demonstrate the efficacy of this method other other multi-shot (In-context learning) prompting schemes and Chain-of-Though (CoT) reasoning with PaLM-2L on various datasets. They also compare against vanilla GPT-4.

**Strengths:**

The methodology is conceptually very clearly explained and motivated. The experiments are extensive, and some useful ablations are carried out. Besides a few points raised below, there is little to critique from the standpoint of methodology or presentation.

The method itself appears to be quite effective, at least for PaLM-2L. It is simple and novel enough to warrant dissemination, given that there is precedent for the publication of prompting methodologies at top-tier conferences (to offer no comment on the scientific merits of this).

**Weaknesses:**

One consistent issue with the paper is the use of incorrect grammar (plurals, subject references etc.) - this issue should be easily remedied through the use of grammar checkers (e.g. Grammarly) or native proof-reading. Notably, in spite of the somewhat jarring errors in English grammar, the sentences are well-structured and the paper has a clear narrative, such that it remains easily comprehensible.

On a related note, the authors consistently use the terms "learn" and "teach" in relation to the step-back question, and the knowledge it provides. This is somewhat confusing, as I don't think any models are fine-tuned etc. to provide this knowledge. Whilst I realise that few-shot prompting is referred to as "in-context *learning*", I would recommend steering clear of this language unless you perform actual updates to the model weights at some stage of the Step-Back Prompting process.

There are two potential methodological weaknesses, which may in fact reflect a misunderstanding on the part of the reviewer.
1) *Baselines might lack useful conditioning in the prompt*. In particular, in section D.2 you state that the baseline prompts only take the question and initial query, whereas Table 11 shows that Step-Back prompting includes the lines e.g. "You are an expert at Physics. You are given a Physics problem". If these are not included in the baseline, then this would appear to be an unfair comparison. Table 15 suggests that the baseline may actually have this information too, so perhaps this is not a concern and section D.2 just omitted this detail.
2) The fact that the methodology is evaluated only on PaLM-2L. I appreciate that GPT-4 calls are not cheap, and Figure 1. provides some evidence for the consistent behaviour of GPT-4 and PaLM-2L. Nonetheless, it is conceivable that this method would not work equally well on other models, and this concern has not been ruled out by the existing experiments.

**Questions:**

Three other things that bear clarifying:
1) How many exemplars are provided for the standard Step-Back experiments (those shown in table 1 etc.). Ablations in Figure 3. suggest it doesn't matter too much, but it would be good to be clear (from Figure 3 one might infer 1 or 5 exemplars are provided, and of course 5 would seem unfair to the baselines)
2) The fact that the step-back question is not generic, but rather already conditions on the context of the dataset (e.g. "What are the physical principles") is worth making more explicit in the methodology. (On a side-note, It would also be interesting to know how a generic prompt such as "Abstract the general principles relevant to this query" would have worked)
3) Why was Step-Back prompting not attempted on GPT-4?

**Details Of Ethics Concerns:**

None of note

---

> ### Author Response · Authors · 2023-11-21
> **Official response to Reviewer XiDg**
>
> Dear Reviewer XiDg,
>
> We sincerely appreciate your comments and feedback which helps to improve our paper. We are grateful for your recommendation to accept our paper! We hope our replies and revisions effectively address your concerns, and please feel free to let us know if you have any further concerns or questions.
>
> Below are our details replies to your comments:
>
> =================================================================
> *One consistent issue with the paper is the use of incorrect grammar (plurals, subject references etc.) - this issue should be easily remedied through the use of grammar checkers (e.g. Grammarly) or native proof-reading. Notably, ..., the sentences are well-structured and the paper has a clear narrative, such that it remains easily comprehensible.*
>
> A: Thanks for the feedback. We have done a thorough grammar and fluency check, and updated the paper accordingly. We appreciate the positive feedback that the paper is “well-structured”, and “easily comprehensible”.
>
> =================================================================
> *On a related note, the authors consistently use the terms "learn" and "teach" in relation to the step-back question, and the knowledge it provides. This is somewhat confusing, as I don't think any models are fine-tuned etc. to provide this knowledge. Whilst I realise that few-shot prompting is referred to as "in-context learning", I would recommend steering clear of this language unless you perform actual updates to the model weights at some stage of the Step-Back Prompting process.*
>
> A: Thanks for the feedback, and you are right that no model is fine-tuned. We have updated the paper to stick with the terminology of "in-context learning" for the term "learn", and have replaced the term "teach" to "demonstrate".
>
> =================================================================
> *Baselines might lack useful conditioning in the prompt. In particular, in section D.2 you state that the baseline prompts only take the question and initial query, whereas Table 11 shows that Step-Back prompting includes the lines e.g. "You are an expert at Physics. You are given a Physics problem". If these are not included in the baseline, then this would appear to be an unfair comparison. Table 15 suggests that the baseline may actually have this information too, so perhaps this is not a concern and section D.2 just omitted this detail.*
>
> A: Thanks for the feedback, and yes, you were correct that we used the same prompting. To make it clear we updated section D.2 to reflect the point that all baselines share the same prompting preamble “You are an expert at Physics. You are given a Physics problem”.
>
> =================================================================
> *The fact that the methodology is evaluated only on PaLM-2L. I appreciate that GPT-4 calls are not cheap, and Figure 1. provides some evidence for the consistent behaviour of GPT-4 and PaLM-2L. Nonetheless, it is conceivable that this method would not work equally well on other models, and this concern has not been ruled out by the existing experiments.*
>
> A: Thanks for the suggestion, we included additional experiments using Llama2-70B and GPT-4 in the updated paper and observed that step-back prompting continues to be competitive against all baseline methods. This demonstrates that our proposed method is model agnostic.
>
> =================================================================
> *How many exemplars are provided for the standard Step-Back experiments (those shown in table 1 etc.). Ablations in Figure 3. suggest it doesn't matter too much, but it would be good to be clear (from Figure 3 one might infer 1 or 5 exemplars are provided, and of course 5 would seem unfair to the baselines)*
>
> A: Thank you for pointing out the potential confusion. We used 1 exemplar for the results throughout the paper, and we explicitly stated this in the updated paper Section 4.3: “Therefore, we use a single exemplar for few-shot prompting throughout the paper except the ablation studies.”
>
> =================================================================
> *The fact that the step-back question is not generic, but rather already conditions on the context of the dataset (e.g. "What are the physical principles") is worth making more explicit in the methodology. (On a side-note, It would also be interesting to know how a generic prompt such as "Abstract the general principles relevant to this query" would have worked)*
>
> A: Thank you for the feedback, and we updated the paper to explicitly state that the step-back question is unique to the task by design in Section 2.
>
> Regarding the point of using a generic prompt, we note that while such a prompt can be helpful for tasks in MMLU, it won’t be applicable to other benchmarks such as TimeQA. For example, a general question of "Abstract the general principles relevant to this query" would not help “Estella Leopold went to which school between Aug 1954 and Nov 1954?"

---

> > ### Comment · Reviewer_XiDg · 2023-11-22
> > **Response to Clarification and Changes**
> >
> > Thank you for your detailed response and clarifications. With the addition of the experiments on Llama2-70B, and clarification on the fairness of the prompting in Section D.2, I am now much more convinced that the methodology's effectiveness is sufficiently substantiated to warrant dissemination.
> >
> > Additionally, the paper reads more fluidly now, though there are still instances of grammatical issues (below two randomly chosen examples) - but these are much less jarring than before and do not hamper the flow of reading. As such, I would say that the paper is sufficiently well-written to meet the standards of publication in Machine Learning.
> >
> > I have updated my scores accordingly.
> >
> > Examples of common mistakes:
> > * "The first step is to show LLMs how to step back in-context learning, and **add:to** derive high-level abstractions such as concepts and principles from the specific example" this is a problematic sentence because there are multiple subjects and actions, and it is not clear which is being referred to at varying points. For example, is "deriving high-level abstractions..." part of "the first step" or is it part of what we are showing LLMs by using in-conext learning? Also, which example is *"the"* specific example? We haven't explicitly referred to an example, so this should be "a given example" etc.. Incorporating these changes would yield a sentence more like "The first step is to show LLMs how to take a step-back by using in-context learning - prompting them to derive high-level abstractions and principles for a given example"
> > * "We conduct a variety of analysis **(analyses)** and find that STEP-BACK PROMPTING has **(provides)** strong performance
> > improvements (up to 36%) over chain of thought (CoT) prompting (Wei et al., 2022b) and **(")**take a
> > deep breathe**(")** (TDB) prompting (Yang et al., 2023)"
> >
> > As you can tell these are fairly pedantic corrections - I just wanted to provide some illustrative examples of minor errors that many authors make; crucially, these don't significantly impact readability in most places though.

---

> > > ### Author Response · Authors · 2023-11-22
> > > **Follow-up response to Reviewer XiDg**
> > >
> > > We are happy to see that our additional experiments and revisions addressed the concerns you had earlier. We are very grateful for your support, and for raising the score.
> > >
> > > We will make changes in the paper for the two corrections pointed out by you, and we will run a native proofreading of the final version.

---

### Official Review · Reviewer_FiKx · 2023-11-01

**Soundness:** 4 excellent
**Presentation:** 3 good
**Contribution:** 4 excellent
**Rating:** 8
**Confidence:** 3

**Summary:**

This work proposes step-back prompting, which prompts the LLM to ask a question about a higher-level concept/principles first. This works as a "retrieval step" which allows it to retrieve relevant facts on which the subsequent reasoning can be grounded.

The model shows good performance on a variety of knowledge-intensive tasks which are typically effectively tackled with RAG methods.

**Strengths:**

The idea of ground reasoning on higher-level abstractions (abstracting away low-level details) is interesting as a principle. It is clear that this kid of reasoning strategy helps on knowledge-intensive tasks, where it helps to retrieve the high-level principles first before proceeding with reasoning.

**Weaknesses:**

While the idea of reasoning from abstract to low level is interesting, the approach explored in the paper is arguably rudimentary - A generic question that asks for a higher-level abstraction only works as a byproduct of the fact that LLM already has near perfect knowledge of such concepts (In Fig 4. the low principle error points to this).

Without a further study of what the kind of abstractions LLM excels at and is still lacking in, my impression is that the method largely functions as better prompt for retrieving relevant facts for knowledge-based (mostly scientific) questions. In my view, the paper would benefit from broadening the exploration of abstraction upon a wider set of tasks.

**Questions:**

1. Is it possible to extend the evaluated tasks to ones involving more broader cases of reasoning, such as GSM8k [1] or bAbi [2]?
That is, do tasks exist in which LLM can fail at deriving the higher-level principle?

2. The paper mentions decomposed prompting in the relate works - It it possible to compare with any such methods other than CoT

3. I'm also curious about the possibility of combining such methods with step-back prompting.

4. Is it possible to study the effect of applying abstraction more than once or in a multi-step manner?



[1] Training Verifiers to Solve Math Word Problems, Cobbe et al., arXiv, 2021
[2] Towards ai-complete question answering: A set of prerequisite toy
tasks. Weston et. al, ICLR, 2016.

---

> ### Author Response · Authors · 2023-11-21
> **Office response to Reviewer FiKx**
>
> Dear Reviewer FiKx,
>
> We sincerely appreciate your comments and feedback which helps to improve our paper. We are grateful for your recommendation to accept our paper! We hope our replies and revisions effectively address your concerns, and please feel free to let us know if you have any further concerns or questions.
>
> Below are our details replies to your comments:
>
> =================================================================
> *Without a further study of what the kind of abstractions LLM excels at and is still lacking in, my impression is that the method largely functions as better prompt for retrieving relevant facts for knowledge-based (mostly scientific) questions. In my view, the paper would benefit from broadening the exploration of abstraction upon a wider set of tasks.*
>
> A: We explored a wide set of reasoning tasks in the paper: STEM, Knowledge QA, Multi-hop Reasoning covering 6 tasks. Nonetheless, we appreciate the suggestion of broadening the exploration of abstraction and have included additional experiments on the GSM8K benchmark, as suggested by the reviewer.
>
> =================================================================
> *Is it possible to extend the evaluated tasks to ones involving more broader cases of reasoning, such as GSM8k or bAbi? That is, do tasks exist in which LLM can fail at deriving the higher-level principle?*
>
> A: Yes, we included additional experiments on GSM8K following the reviewer’s suggestion. For bAbi, we found that models such as GPT-4 have a nearly perfect accuracy (93.3% and 100% ,Tables 2-3 in [1]), and hence there is little headroom for evaluating new techniques on improving bAbi anymore.
>
> Regarding the failure mode of step-back prompting, yes, there do exist tasks in which LLMs can fail at deriving high-level principles and facts. We observe that step-back can have two failure modes discussed in Section 5.3:
> 1. Step-back can fail at deriving the higher-level facts as shown in Figure 5. After step-back, there are still 45% errors due to failure to find the relevant higher-level facts.
> 2. Even with the relevant facts retrieved after step-back, LLMs can still fail the task due to Reasoning errors as in Figure 5: 52% of errors are due to LLMs failing to reason through the facts retrieved from step-back.
>
> [1] GPT-3.5, GPT-4, or BARD? Evaluating LLMs Reasoning Ability in Zero-Shot Setting and Performance Boosting Through Prompts, Espejel, et al., arXiv, 2023
>
> =================================================================
> *The paper mentions decomposed prompting in the relate works - Is it possible to compare with any such methods other than CoT. I'm also curious about the possibility of combining such methods with step-back prompting.*
>
> A: Thanks to the reviewer for the suggestion, we have reached out to the authors of the decomposed prompting to get clarification on how to reproduce their method. We will update the paper once we have a comparison with decomposed prompting.
>
> We appreciate the interesting idea of combining step-back with decomposition, such as using decomposition to help answer the step-back question. We leave this for future explorations.
>
> =================================================================
> *Is it possible to study the effect of applying abstraction more than once or in a multi-step manner?*
>
> A: We believe it is possible. However, in the tasks we studied, we didn’t find it necessary to do multi-step abstractions. Given the potential error compounding effect, doing multi-step step-back may not be helpful if the task complexity doesn’t necessarily require multi-step step-back.

---

> > ### Comment · Reviewer_FiKx · 2023-11-22
> >
> > I appreciate the authors' response and additional experiments. With the additional results, my understanding is clearer that step-back prompting (abstraction to a higher-level principle) is orthogonal to step-by-step reasoning. I have raised my rating to accept.
> > I hope the final version can discuss the differences and necessity of such disparate "reasoning strategies" induced by different prompting methods.

---

> > > ### Author Response · Authors · 2023-11-22
> > > **Follow-up response to Reviewer FiKx**
> > >
> > > We are glad to hear that our responses and revisions addressed the concerns you had earlier, and we greatly appreciate you raising the rating to accept. We will include in the final version a discussion about the differences and necessity of different “reasoning strategies” across prompting methods.

---

### Official Review · Reviewer_xaDx · 2023-11-05

**Soundness:** 4 excellent
**Presentation:** 3 good
**Contribution:** 3 good
**Rating:** 8
**Confidence:** 3

**Summary:**

This paper aims to improve the reasoning ability of large language models, especially for complex tasks that require a large amount of prior knowledge and details. The proposed step-back prompting, first asks a relevant high-level question (called stepback question) and then uses the answer for the following reasoning steps. This stepback question could remind LLM of some principles that are fundamental of the question, and thus improve the reasoning process. Extensive experiments are done on several benchmarks that demonstrate the effectiveness of the step-back prompting compared to the CoT baselines.

**Strengths:**

- Step-back is a reasonable improvement over the existing LLM reasoning prompting strategy. It is especially helpful for tasks that need complex prior information to do reasoning, which broadens LLM's reasoning ability.

- The step-back prompting approach is evaluated with extensive and complementary experiments.

-  The proposed step-back prompting shows significant improvements over several variants of chain-of-thought, including the recent "take a deep breath" on several benchmarks.

**Weaknesses:**

- The step-back prompting is only evaluated on Google's PaLM2. Though it shows significant improvements, it would be difficult for the community to reproduce the results. It would be great to evaluate the proposed prompting approach also on open source LLMs, such as LLaMA2-70B.

-  The step-back question is pretty unique on each benchmark. It seems the specific stepback questions are designed for each benchmark. How to ensure the stepback questions is neat and how to design a perfect stepback question is not clear.

- I feel the stepback question is a special case of the least to most prompting [1], which decomposes complex questions into subquestions and solves them order by order. The stepback questions can also be considered as a subquestion for the following reasoning steps. Can the author further clarify their difference from a principled perspective?

[1] Least-to-Most Prompting Enables Complex Reasoning in Large Language Models, Zhou et al 2023

**Questions:**

Please refer to the previous section.

---

> ### Author Response · Authors · 2023-11-21
> **Official response to Reviewer xaDx**
>
> Dear Reviewer xaDx,
>
> We sincerely appreciate your comments and feedback which helps to improve our paper. We are grateful for your recommendation to accept our paper! We hope our replies and revisions effectively address your concerns, and please feel free to let us know if you have any further concerns or questions.
>
> Below are our detailed replies to your comments:
>
> =================================================================
> *The step-back prompting is only evaluated on Google's PaLM2. Though it shows significant improvements, it would be difficult for the community to reproduce the results. It would be great to evaluate the proposed prompting approach also on open source LLMs, such as LLaMA2-70B.*
>
> A: We thank the reviewer for the suggestion, and include in the update paper additional experiments using LLaMA2-70B and GPT-4 on MMLU demonstrating the model-agonistic aspect of our proposed method.
>
> =================================================================
> *The step-back question is pretty unique on each benchmark. It seems the specific stepback questions are designed for each benchmark. How to ensure the stepback questions is neat and how to design a perfect stepback question is not clear.*
>
> A: We appreciate your question of "how to design a perfect stepback question": it is indeed a very interesting question, as to what is the right abstraction for a particular task. In principle, there could be many ways to do abstraction. For the tasks studied in our papers, we didn’t find it challenging to come up with few-shot demonstrations and make the abstraction idea work.
>
> The step-back question is unique by design in our paper to capture the uniqueness of each benchmark. One potential way of improving the generation of stepback questions could be building a specialized expert fine-tuned model.
>
> =================================================================
> *I feel the stepback question is a special case of the least to most prompting [1], which decomposes complex questions into subquestions and solves them order by order. The stepback questions can also be considered as a subquestion for the following reasoning steps. Can the author further clarify their difference from a principled perspective?*
>
> A: While we agree with the reviewer that the stepback question might look similar to some of the sub questions generated in decomposition methods, here are the key differences:
>
> 1. Step-back questions are abstract and higher level, which are different from least-to-most decompositions that are often low level breakdowns of the original question.
> 2. Abstract questions are often generic in nature with a many-to-one mapping since many questions can have the same abstract questions. This is in contrast to decomposition where there is often a one-to-many mapping since there are multiple decomposed sub-problems necessary to solve a given question.
> 3. Methodology wise, step-back prompting is following a different framework for problem-solving: abstracting to a higher-level to ground reasoning, rather than breaking the problem down into sub-problems in decomposition.
>
> To clarify the differences, we included an illustration example and further explanations in Section 8.2 in the updated paper. We hope this helps clarify the confusion.

---

> > ### Comment · Reviewer_xaDx · 2023-11-23
> > **Post-rebuttal**
> >
> > I appreciate the authors for the further discussion and new experiments in rebuttal. I believe having experiments on open-sourced LLM really benefits the community.  My initial review is high, so I'll just keep the score.

---

### Author Response · Authors · 2023-11-21
**Author Response to all Reviewers**

First, we highly appreciate all the reviewers for their comprehensive reviews, and valuable suggestions and feedback!

**Positive Remarks**
We are glad that our paper received many positive comments, including:
- The acknowledgment of the merits of our proposed Step-Back Prompting method: “especially helpful” [xaDx], “broadens LLM's reasoning ability” [xaDx], “significant improvements over several variants of chain-of-thought” [xaDx], “The idea … is interesting as a principle” [FiKx], “simple and novel enough to warrant dissemination” [XiDg]
- The agreement on the effectiveness of the method: “significant improvements over several variants of chain-of-thought” [xaDx], “good performance on a variety of knowledge-intensive tasks” [FiKx], “quite effective” [XiDg]
- The positive comments on the robustness of the experiments and clarity of our presentation: “evaluated with extensive and complementary experiments” [xaDx], “The experiments are extensive” [XiDg], “conceptually very clearly explained and motivated” [XiDg], “well-structured” [XiDg], “has a clear narrative” [XiDg], and “easily comprehensible” [XiDg].


**Revisions Based on Suggestions and Feedback**
We highly value the constructive feedback and suggestions, and have made the following revisions (highlighted in blue in the updated paper) to improve our paper in terms of both scientific robustness and presentation clarity:
- We have included additional experiments on Llama2-70B and GPT-4 (Table 1), demonstrating the model agnostic aspect of our proposed method.
- We have included additional experiments on the GSM8K benchmark (Appendix A.1).
We have included an example and more explanations in Section 8.2 to illustrate the key differences between Step-Back and Decomposition methods.
- We have addressed other comments on paper writing; e.g., we modified throughout the paper the wording regarding “learn” or “teach LLMs” to “demonstrate to LLMs” to avoid the confusion with model fine-tuning.

We replied to each reviewer in individual responses. We hope our paper revision and responses effectively address your concerns. Please feel free to let us know if you have any further concerns or questions.

Regards,
Authors

---

### Meta-Review · Area_Chair_CivG · 2023-12-13

**Metareview:**

This authors propose step-back prompting, which prompts the LLM to ask a question about a higher-level concept first, answers to which helps the subsequent steps. All reviewers appreciate the novelty and impact of the contribution.

**Justification For Why Not Higher Score:**

NA

**Justification For Why Not Lower Score:**

NA

---

### Decision · Program_Chairs · 2024-01-16

Accept (poster)